# *Aeromonas hydrophila* CobQ is a new type of NAD⁺- and Zn²⁺-independent protein lysine deacetylase

**Yuqian Wang**[1,2†], **Guibin Wang**[1,3†], **Lishan Zhang**[1,4,5], **Qilan Cai**[1,4,5], **Meizhen Lin**[1,4,5], **Dongping Huang**[1,4,5], **Yuyue Xie**[1,4], **Wenxiong Lin**[1,4], **Xiangmin Lin**[1,4,5]*

[1]Fujian Provincial Key Laboratory of Agroecological Processing and Safety Monitoring, Fuzhou, China; [2]Agricultural College, Anhui Science and Technology University, Chuzhou, China; [3]State Key Laboratory of Proteomics, Beijing Proteome Research Center, National Center for Protein Sciences (Beijing), Beijing Institute of Lifeomics, Beijing, China; [4]Key Laboratory of Crop Ecology and Molecular Physiology (Fujian Agriculture and Forestry University), Fujian Province University, Fuzhou, China; [5]Key Laboratory of Marine Biotechnology of Fujian Province, Institute of Oceanology, Fujian Agriculture and Forestry University, Fuzhou, China

**\*For correspondence:**
xiangmin@fafu.edu.cn

†These authors contributed equally to this work

**Competing interest:** The authors declare that no competing interests exist.

## eLife Assessment

In this **valuable** study, the authors studied a novel Zn2+- and NAD+-independent KDAC protein, AhCobQ, in Aeromonas hydrophila, which lacks homology with eukaryotic counterparts, thus underscoring its unique evolutionary trajectory within the bacterial domain. They attempt to demonstrate deacetylase activity, however, whilst the revised manuscript has been improved, significant aspects of the data are still **incomplete** and require further refinement. The work will be of interest to microbiologists studying metabolism and post-translational modifications.

**Abstract** Protein Nᵋ-lysine acetylation (Kac) modifications play crucial roles in diverse physiological and pathological functions in cells. In prokaryotic cells, there are only two types of lysine deacetylases (KDACs) that are Zn²⁺- or NAD⁺-dependent. In this study, we reported a protein, AhCobQ, in *Aeromonas hydrophila* ATCC 7966 that presents NAD⁺- and Zn²⁺-independent KDAC activity. Furthermore, its KDAC activity is located in an unidentified domain (from 195 to 245 aa). Interestingly, AhCobQ has no homology with current known KDACs, and no homologous protein was found in eukaryotic cells. A protein substrate analysis showed that AhCobQ has specific protein substrates in common with other known KDACs, indicating that these KDACs can dynamically co-regulate the states of Kac proteins. Microbiological methods employed in this study affirmed AhCobQ's positive regulation of isocitrate dehydrogenase (ICD) enzymatic activity at the K388 site, implicating AhCobQ in the modulation of bacterial enzymatic activities. In summary, our findings present compelling evidence that AhCobQ represents a distinctive type of KDAC with significant roles in bacterial biological functions.

## Introduction

In recent years, due to the development of specific antibody enrichment technologies and high-resolution mass spectrometry, the research on protein post-translational modifications (PTMs) is increasing dramatically, and it provides a new platform for the study of bacterial function and

regulation. PTMs refer to small molecular moieties that dynamically affect the structure and function of proteins through chemical modification of specific amino acid residues, so as to regulate many important biological processes, including cell metabolism, DNA transcription and replication, host infection, and stress resistance, therefore, play an important role in almost all physiological functions (*Carabetta et al., 2019*). Among them, the Kac modification on protein is widely found in bacteria and participates in almost all key biological processes in bacterial life activities and is attracting the attention of microbiologists (*Yu et al., 2023*).

Although there are many reports on the identification of acetylation modification patterns of bacterial protein lysine residues in recent years, two fundamental questions have not been solved: how do these acetylation modifications arise (acetylation modifiers; writers) and how are they removed (deacetylation modifiers; erasers) (*Wang and Cole, 2020a*). On the one hand, acetyl groups can be transferred to the target protein by acetyltransferases or non-enzymatic (chemical) mechanisms *VanDrisse and Escalante-Semerena, 2019*; on the other hand, proteins need to remove acetyl groups from modified proteins to reversibly regulate the function of target proteins. At present, the research on the functions and types of lysine deacetylases (KDACs) has been well-documented in eukaryotes, and most of them are histone deacetylases (HDACs). There are currently at least 18 types of HDACs in four classes (I–IV) in mammals and four plant-specific histone deacetylase 2 (HD2) members in the plant (*Yoshida et al., 2017*; *Chen et al., 2020*). Of these, HDAC types I, II, and IV are $Zn^{2+}$-dependent, while type III deacetylases are $NAD^+$-dependent sirtuins. Moreover, sirtuins can be divided into seven types (SIRT 1–7), which are distributed in different compartments of eukaryotic cells and perform important biological functions (*Narita et al., 2019*). Besides these, ABHD14B, a novel KDAC, was reported to deacetylate non-histone substrates and influence glucose metabolism in mammals (*Rajendran et al., 2022*; *Rajendran et al., 2020*).

The currently known KDACs in bacteria include $Zn^{2+}$-dependent RpLdaA or Kdac1 (homologous to eukaryotic type II KDACs), $Zn^{2+}$-dependent AcuC (homologous to type I), and $NAD^+$-dependent sirtuin (homologous to type III) family proteins (*Watson and Christianson, 2023*). Sirtuins, represented by *Escherichia coli* CobB, are the most widely distributed KDACs in many bacterial species and are involved in the regulation of a variety of important physiological functions (*Burckhardt et al., 2019*; *Liu et al., 2018*; *Wang et al., 2020b*). However, since the first report of the deacetylase function of CobB in *Salmonella typhimurium* in 1998, only two types (three classes) of KDACs have been confirmed in bacteria (*Tsang and Escalante-Semerena, 1998*; *Macek et al., 2019*). The reasons that hinder new KDAC discovery in bacteria may be the following: (1) due to the in-depth study of KDACs in eukaryotes, almost all the current research on bacterial deacetylases are based on the characteristics of their homologous proteins in eukaryotes, but less attention is paid to bacteria-specific KDACs, and (2) the deacetylation process between protein substrates and KDACs may be a transient and reversible weak interaction. Moreover, there are thousands of lysine acetylation modification sites in cellular proteins (*Chen et al., 2022*); so, it is difficult to speculate the characteristics of their specific binding deacetylase, which makes the efficiency of enriching specific KDACs by common protein-protein interaction methods, such as co-immunoprecipitation and pull-down technologies, challenging. As a result, the identification and mechanism of new deacetylases in prokaryotic cells are lagging. Therefore, the screening and identification of new deacetylases, especially for new types of KDACs that may not depend on $NAD^+$ or $Zn^{2+}$, have been a bottleneck limiting the research on the regulatory mechanisms of bacterial acetylation/deacetylation modifications.

In this study, we fortuitously identified the protein AhCobQ in *Aeromonas hydrophila* ATCC 7966 (UniProt ID A0KI27, gene ID *AHA_1389*) that presents $NAD^+$- and $Zn^{2+}$-independent protein lysine deacetylase activity. Interestingly, although this protein is described as a CobQ/CobB/MinD/ParA family protein in the UniProt database, it is homologous with cobyric acid synthase in other bacterial species but shares no homology with CobB sirtuin family proteins. In addition to deacetylating lysine-acetylated sites common to AhCobB (UniProt ID A0KKN9, gene ID *AHA_2317*) sirtuin and another deacetylase AhAcuC (UniProt ID A0KLZ2, gene ID *AHA_2788*) in *A. hydrophila*, we also confirmed that AhCobQ deacetylated specific lysine-acetylated sites as did other KDACs, even in the same protein. In general, our results demonstrated a new type of bacterial KDAC, which shares no homology with currently known lysine deacetylases, including those in eukaryotic cells, and extends our understanding of the complicated regulatory mechanism of bacterial lysine acetylation modifications.

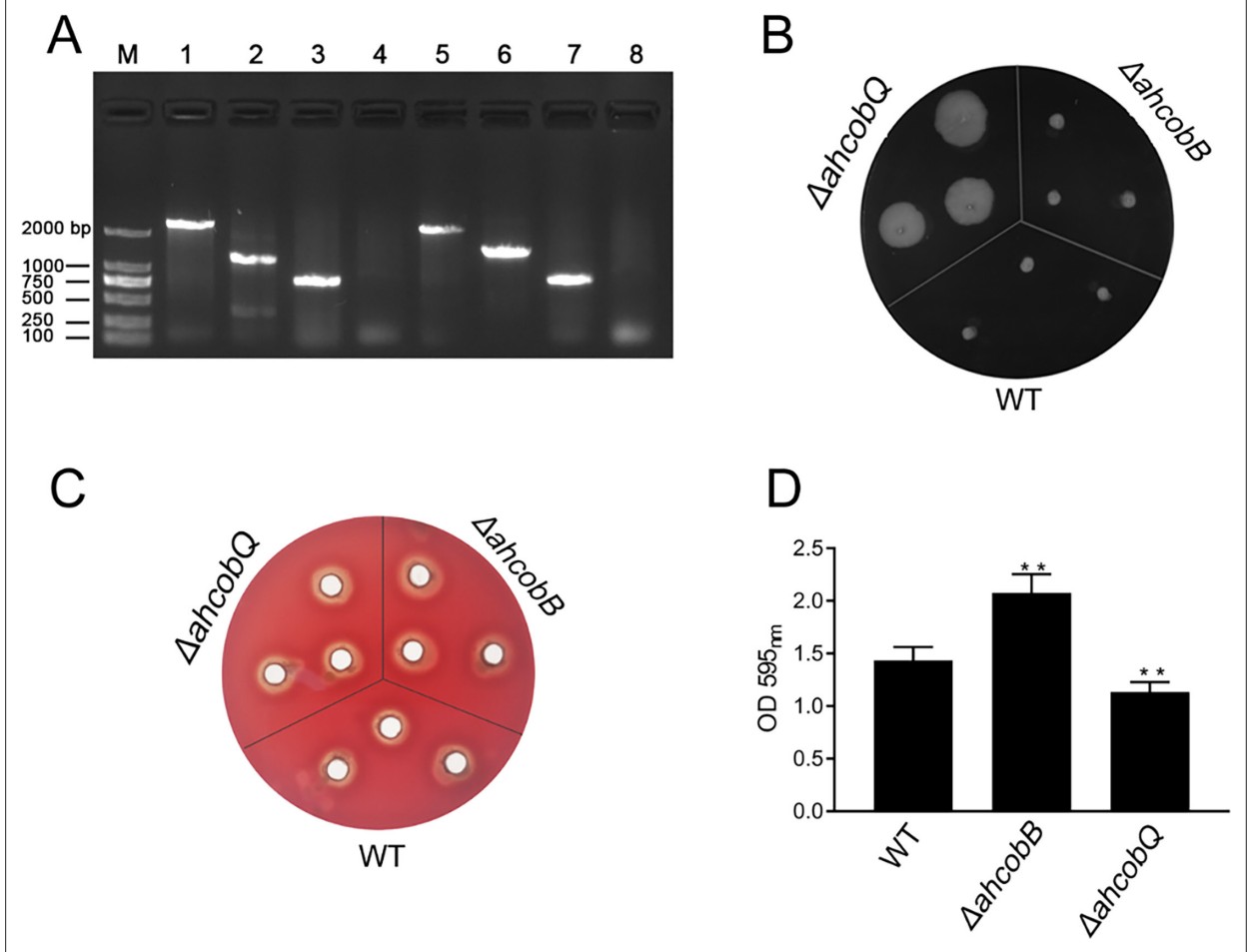

**Figure 1.** Phenotypes of ΔahcobB and ΔahcobQ strains. (**A**) Construction of *ahcobB*- and *ahcobQ*-defective strains. M: Marker; Lanes 1 and 3: PCR products of wild-type (WT) using the P7/P8 primer pairs of *ahcobB* and *ahcobQ*, respectively; Lanes 2 and 6: PCR products of ΔahcobB and ΔahcobQ, respectively, using P7/P8 primer pairs; Lanes 5 and 7: PCR products of WT using P5/P6 primer pairs of *ahcobB* and *ahcobQ*, respectively; Lanes 4 and 8: PCR products of ΔahcobB and ΔahcobQ, respectively, using the P5/P6 primer pairs. (**B**) Bacterial migration ability; (**C**) hemolytic activity; (**D**) histogram of the biofilm formation ability (OD 595 nm). \*\*p<0.005.

The online version of this article includes the following source data and figure supplement(s) for figure 1:

**Source data 1.** PDF file containing original PCR for *Figure 1A*, indicating the relevant bands and treatments.

**Source data 2.** Original files for PCR analysis displayed in *Figure 3D*.

**Figure supplement 1.** Bacterial swarming ability of ΔahcobQ, wild-type (WT), and *ahcobQ* complement strain.

## Results

### The physiological phenotypes of AhCobQ are different from those of AhCobB

This project proceeded from the misunderstanding that AhCobQ and AhCobB may be homologous proteins because AhCobQ is annotated as a CobQ/CobB/MinD/ParA family protein in the UniProt database. Therefore, we constructed *ahcobB* and *ahcobQ* mutant strains (*Figure 1A*) and compared their general physiological phenotypes. When compared to the wild-type strain (WT), the deletion of *ahcobB* did not affect the hemolytic or swarming ability of *A. hydrophila*. Notably, ΔahcobQ did not affect the hemolytic ability either but significantly increased the swarming ability (*Figure 1B, C* and *Figure 1—figure supplement 1*). Most interestingly, ΔahcobB sharply increased, while ΔahcobQ significantly decreased, the bacterial biofilm formation ability (*Figure 1D*). These results indicated that CobB and CobQ in *A. hydrophila* may play different roles in biological functions.

## AhCobQ and AhCobB are not homologous proteins

We then further investigated the evolutionary relationship between AhCobQ and AhCobB. To our surprise, we found that the proteins belong to different protein families. AhCobB (UniProt ID A0KKN9) belongs to a well-known $NAD^+$-dependent protein deacylase sirtuin family and shares a highly homologous SIR2 domain with CobB sirtuins in many bacterial species, such as *E. coli* (NPD_ECOLI), *S. typhimurium* (NPD_SALTY), *Pseudomonas nitroreducens* (A0A246F352_9PSED), and even seven types of sirtuins in humans (*Figure 2A*). However, AhCobQ (UniProt ID A0KI27) shares an AAA_31 ATPase domain with several homologous proteins, such as cobalamin biosynthesis protein CobQ in *P. nitroreducens* (A0A246FC44_9PSED), probable plasmid partitioning protein in *P. aeruginosa* (G3XCW7_PSEAE), and chromosome partitioning ATPase in *S. typhimurium* (A0A0F6B556_SALT1). Moreover, AhCobQ also shares homology with Soj protein (A0KQY7_AERHH) and cell division inhibitor MinD (A0KK56_AERHH) in *A. hydrophila* as well (*Figure 2B* and *Figure 2—figure supplement 1*). Interestingly, this protein was not found to have homologous proteins in humans or mice species, indicating the significant evolutionary difference between AhCobQ and sirtuins.

## AhCobQ is an NAD- and $Zn^{2+}$-independent lysine deacetylase

Dot blot and western blot experiments showed that the deletion of *ahcobQ* or *ahcobB* both significantly increased the whole protein lysine acetylation levels in *A. hydrophila* cells (*Figure 2C and D*), which aroused our interest as to whether AhCobQ is a lysine deacetylase or not. To test this possibility, we purified AhCobQ, AhAcuC, and AhCobB proteins and incubated them in vitro with lysine acetylated-bovine serum albumin (Kac-BSA) as a protein substrate with and without $NAD^+$ or $Zn^{2+}$ treatment (*Figure 2—figure supplement 2* and *Figure 2—figure supplement 3*). As a positive control, AhCobB and AhAcuC significantly deacetylated Kac-BSA in the presence of $NAD^+$ or $Zn2^+$ buffer, which met our expectations. However, AhCobQ clearly deacetylated Kac-BSA regardless of whether $NAD^+$ or $Zn2^+$ were present (*Figure 2E and G*). Moreover, the KDAC activity of AhCobB was inhibited when the sirtuin inhibitor nicotinamide (NAM) was added, but this did not affect the deacetylase activity of AhCobQ (*Figure 2F*). Furthermore, the introduction of the metal ion chelator, EDTA, had no discernible effect on the deacetylase activity of AhCobQ (*Figure 2—figure supplement 4*). These results strongly indicate that AhCobQ is an $NAD^+$- and $Zn^{2+}$-independent protein deacetylase. Additionally, since AhCobQ has an ATPase domain at the N-terminal, we also tested its deacetylase activity with or without ATP (*Figure 2H* and *Figure 2—figure supplement 5*). The results demonstrated that the KDAC activity of AhCobQ was ATP-independent, suggesting its enzymatic activity was downstream of the ATPase domain.

We further digested Kac-BSA to peptides after AhCobQ incubation and then subjected the peptides to LC MS/MS to identify their acetylated sites (*Supplementary file 1a*). The results showed that the MS intensities of at least 15 Kac peptides in BSA were significantly decreased after AhCobQ treatment. To further validate the KDAC characteristics of AhCobQ and exclude the possibility that AhCobQ may hydrolyze Kac peptides or switch the lysine acetylation to other unknown PTMs, we selected and synthetized an acetylated peptide sequence (YNGVFQECCQAEDK$^{ac}$GACLLPK) with its unacetylated peptide as negative control and subjected them to additional deacetylation assays. The synthetized acetylated (STD*$^{Kac}$) and deacetylated (STD) peptides were incubated with AhCobB and AhCobQ and then identified by LC MS/MS. As shown in *Figure 3*, the masses of the STD*$^{Kac}$ and STD peptides were 786.35538 (3+) and 772.35175 (3+), respectively. When the STD*$^{Kac}$ peptide was incubated with AhCobB or AhCobQ, the corresponding retention time in the C18 chromatography and the MS1 peak of the deacetylated peptide were found in both treatments (*Figure 3A and B*). Moreover, the deacetylated and acetylated peptides were further validated in MS2 by LC MS/MS (*Figure 3C and D*). These results indicate that AhCobQ is not a hydrolase that hydrolyzes target proteins but likely a protein deacetylase similar to AhCobB.

## The deacetylase activity of AhCobQ is in an unidentified domain

Because AhCobQ has a conservative N-terminal AAA family ATPase domain (1–179 aa) and its KDAC activity was not affected by ATP, we then asked whether this domain has deacetylase activity. We divided and purified recombinant AhCobQ protein into two truncated protein fragments of 1–179 aa and 179–264 aa and then tested their deacetylase activities. To our surprise, the 179–264 aa protein fragment but not the 1–179 aa fragment showed $NAD^+$- and $Zn^{2+}$-independent deacetylase activity

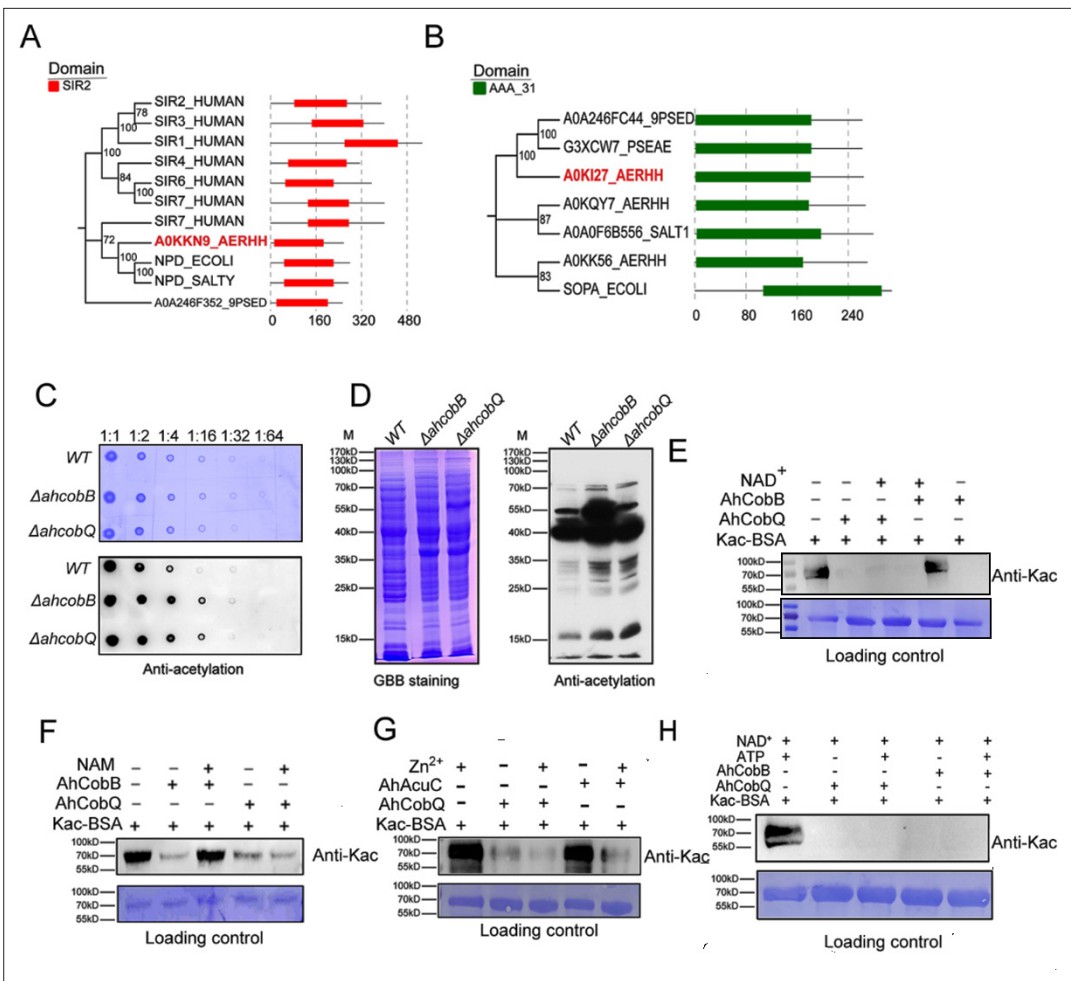

**Figure 2.** Homology comparison and deacetylase activity assay of AhCobQ and AhCobB. (**A, B**) Homologous and conserved domain analysis of AhCobQ and AhCobB family proteins. (**C, D**) Dot blot and western blot verified the whole cell protein Kac level among WT, ΔahcobB, and ΔahcobQ strains. (**E–H**) Effect of $NAD^+$, NAM, $Zn^{2+}$, and ATP on KDAC enzymatic activity of AhCobQ.

The online version of this article includes the following source data and figure supplement(s) for figure 2:

**Source data 1.** PDF file containing original western blots for *Figure 2C–H*, indicating the relevant bands and treatments.

**Source data 2.** Original files for western blot analysis displayed in *Figure 2C–H*.

**Figure supplement 1.** Comparison of homology of AhCobQ protein sequences by using BLAST.

**Figure supplement 2.** Characteristics of overexpressed and purified recombinant proteins AhCobQ, AhCobB, and AhAcuC.

**Figure supplement 2—source data 1.** PDF file containing original PCR and SDS-PAGE for *Figure 2—figure supplement 2*, indicating the relevant bands.

**Figure supplement 2—source data 2.** Original files for PCR and SDS-PAGE analysis displayed in *Figure 2— figure supplement 2*.

**Figure supplement 3.** In vitro acetylated BSA (Kac-BSA) was verified by western blot.

**Figure supplement 3—source data 1.** PDF file containing western blots for *Figure 2—figure supplement 3*, indicating the relevant bands and treatments.

**Figure supplement 3—source data 2.** Original files for western blots analysis displayed in *Figure 2—figure supplement 3*.

**Figure supplement 4.** In vitro deacetylase activity assay of different recombinant AhCobQ proteins in the presence and absence of EDTA utilizing Kac-BSA as substrate.

*Figure 2 continued on next page*

*Figure 2 continued*

**Figure supplement 4—source data 1.** PDF file containing original western blots for *Figure 2—figure supplement 4*, indicating the relevant bands and treatments.

**Figure supplement 4—source data 2.** Original files for western blots analysis displayed in *Figure 2—figure supplement 4*.

**Figure supplement 5.** The lysine deacetylase activity of AhCobQ didn't be affected by 0.5 mM ATP at different incubation times.

**Figure supplement 5—source data 1.** PDF file containing original western blots for *Figure 2—figure supplement 5*, indicating the relevant bands and treatments.

**Figure supplement 5—source data 2.** Original files for western blots analysis displayed in *Figure 2—figure supplement 5*.

(*Figure 4A*). To better understand the deacetylase domain of AhCobQ, a total of 12 truncated protein fragments were constructed and purified, and their deacetylase activities were evaluated (*Figure 4—figure supplement 1* and *Figure 4A*). As shown in *Figure 4*, the 1–179 aa, 200–255 aa, and 189–240 aa shortened proteins disrupted the deacetylase activity of AhCobQ to some extent, whereas the 179–264 aa, 189–264 aa, 199–264 aa, 195–264 aa, 195–255 aa, 189–245 aa, 189–250 aa, 189–255 aa, and 189–264 aa shortened proteins did not, indicating that the deacetylase activity may be in the 195–245 aa range. Interestingly, although the full amino acid sequence of AhCobQ (1–254 aa) is homologous with Soj protein (A0KQYT), site-determining protein (A0KI19 and A0KK56), and ParA family protein (A0KQG8) in *A. hydrophila*, the KDAC activity domain (195–245 aa) does not share homology with these proteins, which suggests the different biological roles among them. Moreover, AhCobQ (195–245 aa) also shares homology with several prokaryotic species, such as *Aeromonas sp.*, *Aliivibrio sp.*, *Enterovibri sp.*, *Oceanimonas sp.*, *Photobacterium sp.*, and *Vibrio sp.*, most of which are marine prokaryotes based on a PSI-BLAST program searching, but the search failed to find homologous proteins in eukaryotic cells (*Figure 5*). Therefore, the 195–245 aa shortened protein of AhCobQ may represent a new domain that exhibits $NAD^+$- and $Zn^{2+}$-independent deacetylase activity.

## AhCobQ, AhCobB, and AhAcuC deacetylate different acetylated proteins and sites

We then identified the proteins that were substrates of AhCobQ and analyzed the substrate differences among AhCobQ, AhCobB, and a predicted $Zn^{2+}$-dependent deacetylase AhAcuC of *A. hydrophila* that is homologous to type I KDACs in eukaryotes. A quantitative lysine acetylome was generated to compare the differential expression of lysine-acetylated peptides among *ahcobQ*, *ahcobB*, and *ahacuC* gene-deficient strains and their WT strain (*Figure 6*, *Figure 6—figure supplement 1*, *Figure 6—figure supplement 2*, *Figure 6—figure supplement 3* and *Supplementary file 1b*). We only focused on the upregulated lysine-acetylated peptides between the mutant and WT strains. When compared to WT, the deletion of *ahcobQ*, *ahcobB*, and *ahacuC* led to increased abundances at 52 Kac sites (corresponding to 47 Kac proteins), 370 Kac sites (291 proteins), and 197 Kac sites (176 proteins), respectively. Moreover, there were 19 increased Kac peptides common between the Δ*ahcobQ* and Δ*ahacuC* strains, 10 increased Kac peptides common between the Δ*ahcobQ* and Δ*ahcobB* strains, 69 increased Kac peptides common between the Δ*ahcobB* and Δ*ahacuC* strains, and four peptides common among the Δ*ahcobQ*, Δ*ahcobB*, and Δ*ahacuC* strains (*Figure 6A*). We also found that there were only 531 Kac peptides directly regulated by these three deacetylases, which is only a small proportion of the total Kac peptides (accounting for 10.28% of all 3163 Kac peptides identified in the current study), indicating the presence of other unknown KDACs in this bacterium.

A KEGG pathway analysis showed that biosynthesis of amino acids and antibiotics was enriched in the unique upregulated Kac proteins in the Δ*ahcobQ* strain (*Figure 6B*). Furthermore, five pathways, namely, pyrimidine metabolism, metabolic pathways, purine metabolism, pyruvate metabolism, and RNA polymerase, were enriched in upregulated Kac proteins under AhCobB-regulation. Of these three deacetylase mutants, Δ*ahacuC* enriched the most metabolic pathways, although Δ*ahcobB* regulated more Kac sites. Carbon metabolism, biosynthesis of antibiotics, and glycolysis/gluconeogenesis pathways were the top three enrichment pathways in the unique 119 Kac sites regulated by AhAcuC. Phosphotransferase system (PTS), microbial metabolism in diverse environments, and metabolic

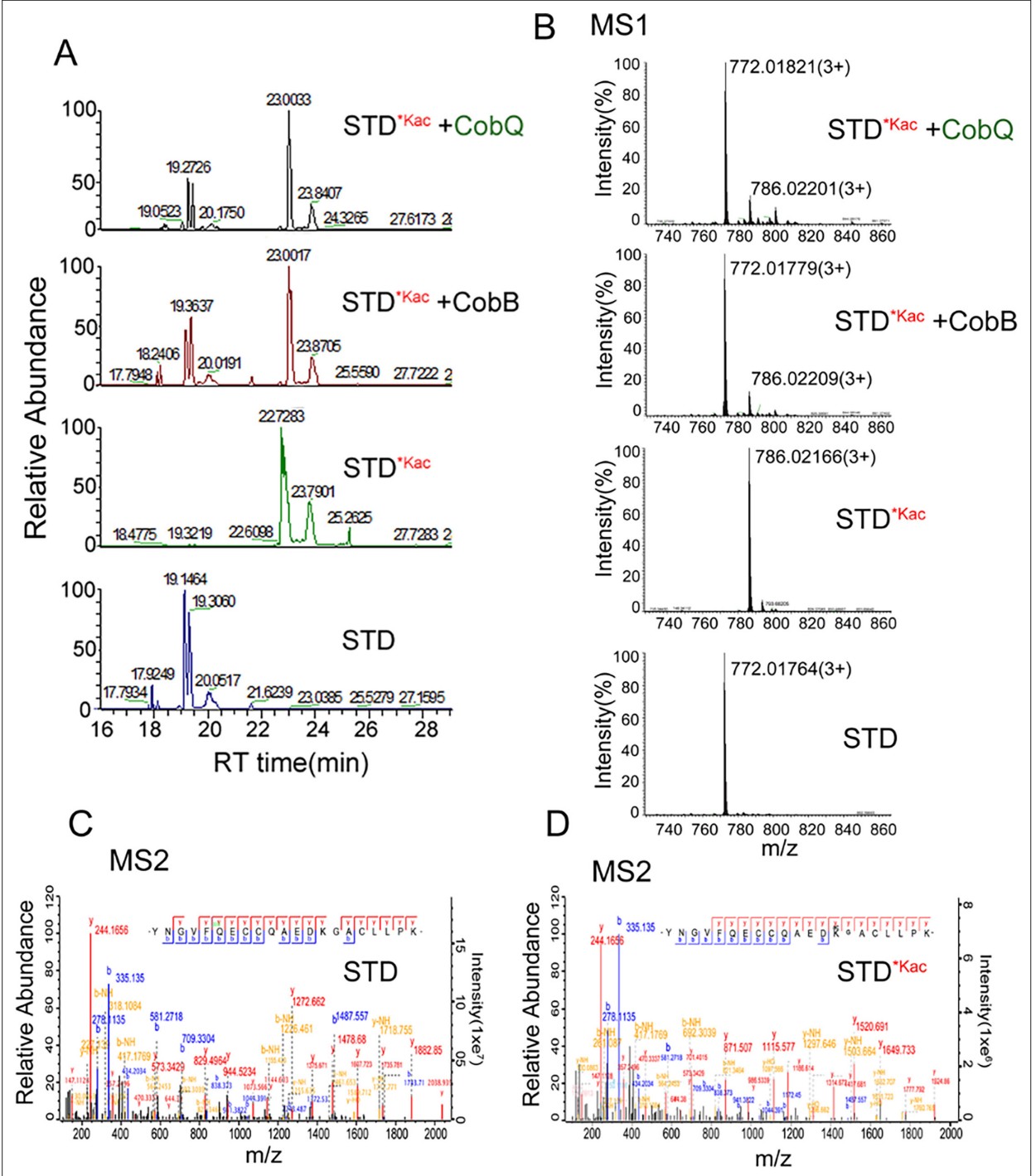

**Figure 3.** Validation of the KDAC activity of AhCobQ by LC-MS/MS. (**A**) Retention time of synthetic peptides YNGVFQECCQAEDKGACLLPK (STD) and YNGVFQECCQAEDKᵃᶜGACL-L-PK (STD*Kac) with or without AhCobB and AhCoQ treatment; (**B**) MS1 spectrum of synthetic peptides STD and STD*Kac with or without AhCobB and AhCobQ treatment; (**C, D**) MS2 spectrum of acetylated and unmodified peptides.

pathways were enriched in the upregulated Kac proteins common to the ΔahacuC and ΔahcobB mutants. In general, although there are different pathways involved in the activity of different deacetylases, most of the Kac proteins regulated by these KDACs were enzymes and normally had important roles in metabolic pathways.

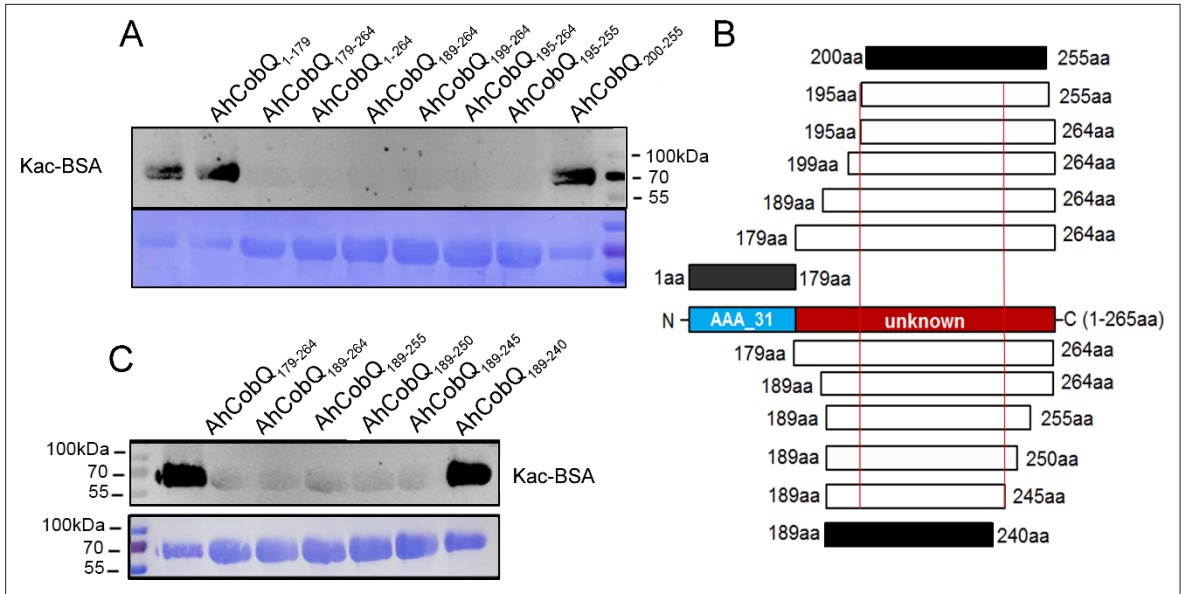

**Figure 4.** Lysine deacetylase activity of AhCobQ truncated proteins. Western blot analysis of the lysine acetylases and deacetylases (KDAC) activity of (**A**) AhCobQ$_{1-179}$, AhCobQ$_{179-264}$, AhCobQ$_{1-264}$, AhCobQ$_{189-264}$, AhCobQ$_{199-264}$, AhCobQ$_{195-264}$, AhCobQ$_{195-255}$, and AhCobQ$_{200-255}$; (**C**) AhCobQ$_{179-264}$, AhCobQ$_{189-264}$, AhCobQ$_{189-255}$, AhCobQ$_{189-250}$, AhCobQ$_{189-245}$, and AhCobQ$_{189-240}$ truncated proteins treated with Kac-BSA. The first lane represents Kac-BSA without AhCobQ truncated proteins. The upper part of the figure shows the WB results of the Kac level of truncated proteins with Kac-BSA, and the lower part shows the PVDF membrane R350 staining for the loading amount control. (**B**) Design of the series of truncated proteins and the summary of the WB results in this experiment. Blue indicates the AAA_31 domain (1–179 aa), and red indicates the unknown range (180–264 aa) in AhCobQ. Black and white indicate the positive and negative WB results, respectively, in this study.

The online version of this article includes the following source data and figure supplement(s) for figure 4:

**Source data 1.** PDF file containing original western blots for *Figure 4A and C*, indicating the relevant bands and treatments.

**Source data 2.** Original files for western blots analysis displayed in *Figure 4A and C*.

**Figure supplement 1.** Characteristics of overexpressed and purified recombinant AhCobQ truncated proteins.

**Figure supplement 1—source data 1.** PDF file containing original PCR and SDS-PAGE for *Figure 4—figure supplement 1*, indicating the relevant bands.

**Figure supplement 1—source data 2.** Original files for PCR and SDS-PAGE analysis displayed in *Figure 4—figure supplement 1*.

## Validation of the specificity of Kac proteins and sites by deacetylases

We further validated the specificity of Kac proteins and the sites regulated by these deacetylases. A total of three target proteins (SUN, ENO, and ArcA-2) were selected for validation due to their significantly upregulated Kac levels in the Δ*ahcobQ* strain, potentially indicating their roles as substrates for AhCobQ. These proteins were first cloned and purified in *E. coli*. The anti-Kac western blot showed that these purified proteins did not have lysine acetylation signals, which may be because of the heterologous expression in an engineered bacterial strain. This was ideal for excluding signal interference from the background (*Figure 7—figure supplement 1*). Next, site-specific lysine acetylation of the target protein substrates was constructed using a two-plasmid-based system of genetically encoded N$^{\varepsilon}$-acetyllysine, which was purified and incubated with AhCobQ, AhCobB, and AhAcuC for western blot analysis against anti-lysine acetylation antibodies (*Figure 7—figure supplement 2*). As shown in *Figure 7*, when compared to untreated Kac proteins, the acetylation level of three Kac sites (K103 and K148 in SUN, K195 in ENO) that were uniquely regulated by AhCobQ according to quantitative acetylome analysis was significantly reduced after incubation with AhcobQ but unchanged after incubation with AhCobB or AhAcuC. Moreover, K26 in ArcA-2 was reduced in both AhCobB and AhCobQ incubations. These results were consistent with our quantitative acetylome results.

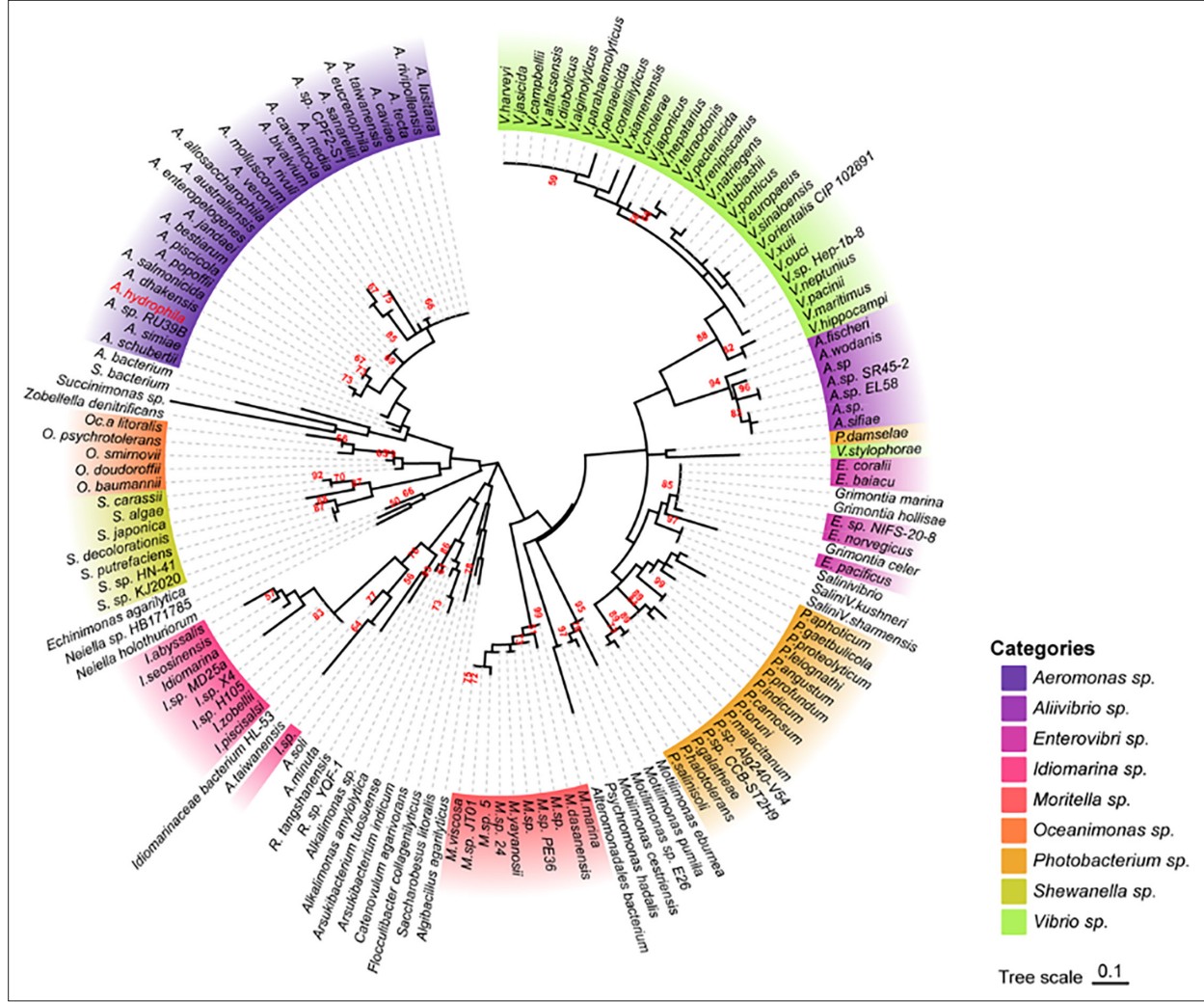

**Figure 5.** Phylogenetic tree of the AhCobQ lysine acetylases and deacetylases (KDAC) activity domain (195–245 aa). Different colors indicate different branches, and *Aeromonas hydrophila spp.* is highlighted in red.

## AhCobQ positively regulates the enzymatic activity of isocitrate dehydrogenase (ICD) at K388 site

To further understand the effect of AhCobQ on bacterial biological function, an AhCobQ substrate, *A. hydrophila* ICD was cloned and purified in *E. coli* (**Figure 7—figure supplement 2**). Western blot assay showed that this protein didn't display lysine acetylation modification in *E. coli* (**Figure 8A**). Therefore, site-specific lysine acetylation was constructed at the K388 site of ICD for further enzymatic activity determination. As shown in **Figure 8B**, AhCobQ can significantly deacetylated ICD at K388, while the other two deacetylases, AhCobB and AhAcuC didn't. This result is consistent with the proteomics conclusions. When compared with unKac ICD protein, the Kac at K388 of ICD significantly decreased the product of NADPH with the addition of NADP⁺ substrate, which indicates the Kac at K388 negatively regulates the enzymatic activity of ICD. Moreover, the adding of AhCobQ significantly increased the enzymatic activity of K388 acetylated ICD and quickly reached the platform period in 10 min, while AhCobB and AhAcuC didn't affect ICD enzymatic activity (**Figure 8C and D**). These results indicate that AhCobQ positively regulates the enzymatic activity of isocitrate dehydrogenase by deacetylating the ICD protein at the K388 site.

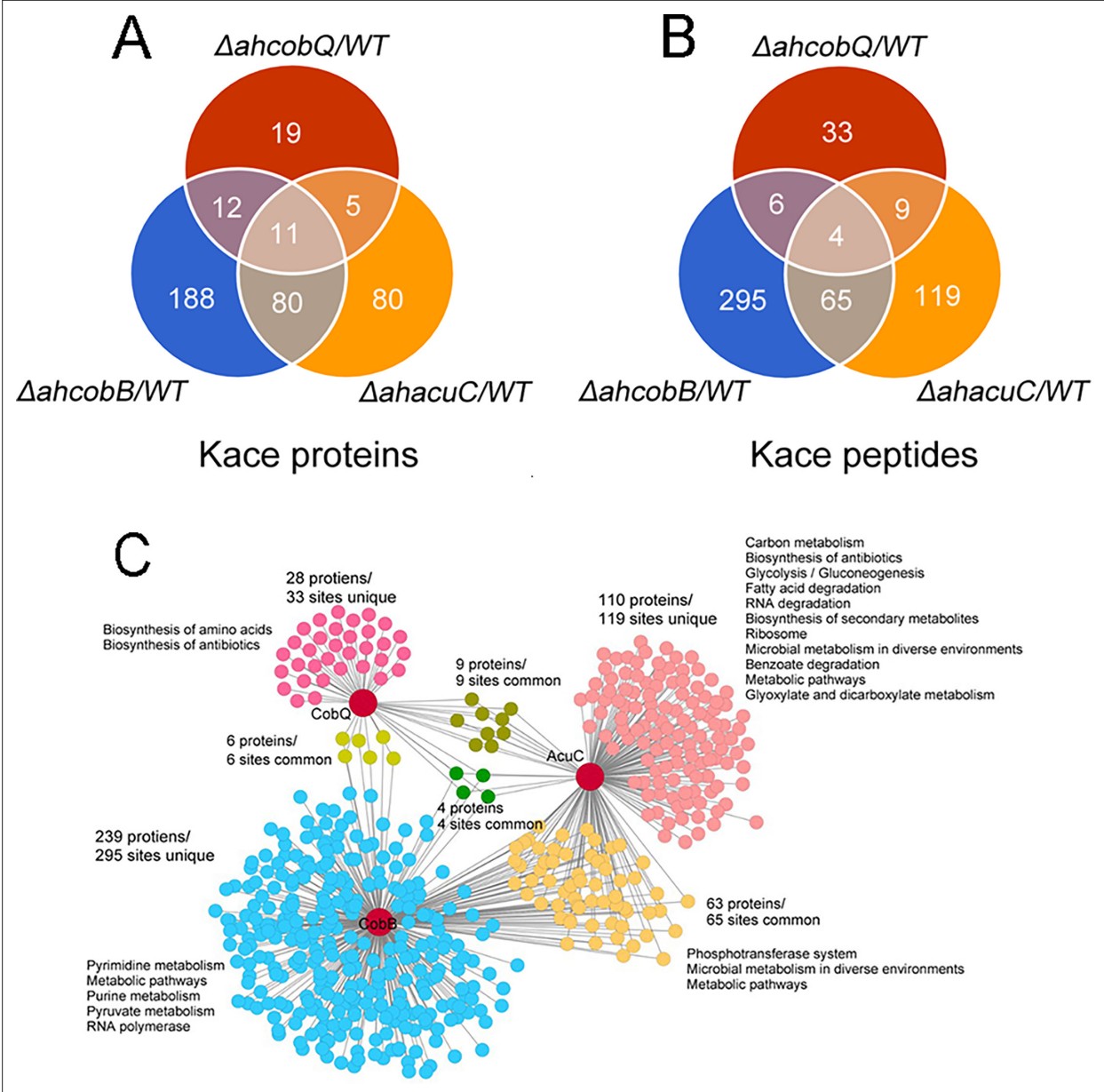

**Figure 6.** Proteomics analysis of upregulated proteins in Δ*ahcobB*, Δ*ahcobQ*, Δ*ahacuC*, and wild-type (WT) strains. (**A**) Venn diagrams visualized the overlapped upregulated $N^\varepsilon$-lysine acetylation (Kac) proteins and Kac peptides among the Δ*ahcobB*, Δ*ahcobQ*, Δ*ahacuC*, and WT strains. The number of acetylated proteins and peptides (sites) are shown in the figure. (**B**) Kyoto Encyclopedia of Genes and Genomes (KEGG) metabolic pathway enrichment analysis of upregulated Kac proteins and peptides by the three lysine acetylases and deacetylases (KDACs). Unique and overlapped Kac proteins and peptides (sites) are presented in different colors.

The online version of this article includes the following source data and figure supplement(s) for figure 6:

**Figure supplement 1.** Construction of *A*.

**Figure supplement 1—source data 1.** PDF file containing original PCR for *Figure 6—figure supplement 1*, indicating the relevant bands.

**Figure supplement 1—source data 2.** Original files for PCR analysis displayed in *Figure 6—figure supplement 1*.

**Figure supplement 2.** Scatter plot of Pearson's correlation coefficients of label-free quantification (LFQ) intensity among each group and their biological replicates in the quantitative acetylome.

**Figure supplement 3.** Quantitative acetylome analysis among the Δ*ahcobQ*, Δ*ahcobB*, Δ*ahacuC*, and wild-type (WT) strains.

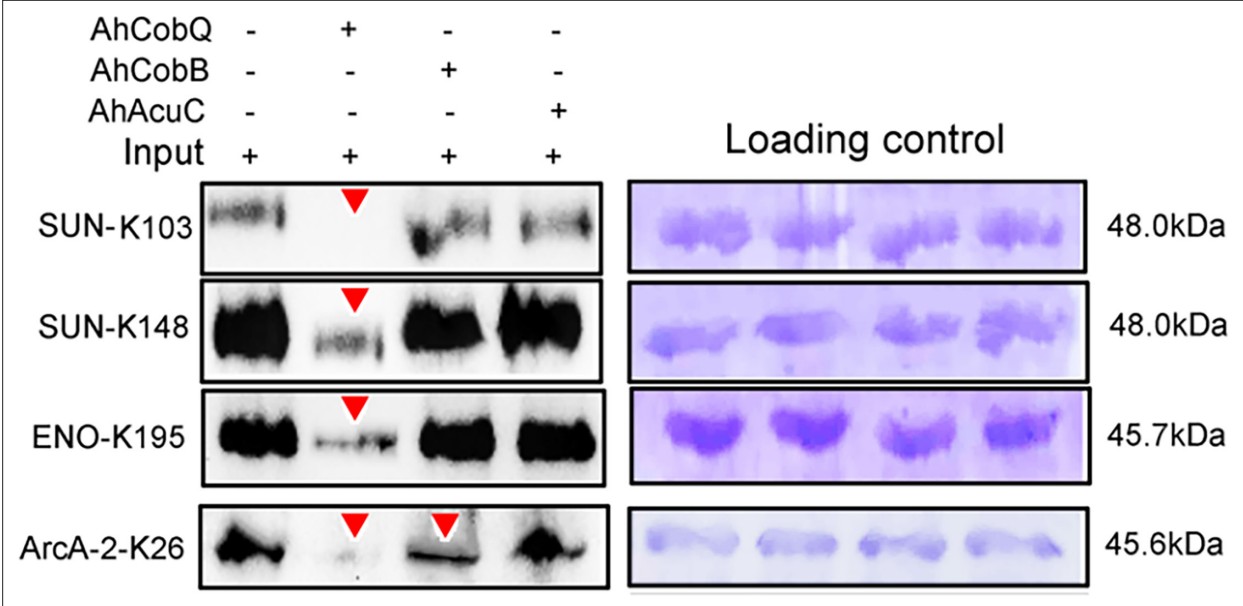

**Figure 7.** Western blot validation of the site-specific $N^\varepsilon$-lysine acetylation (Kac) protein substrates regulated by the three KDACs.

The online version of this article includes the following source data and figure supplement(s) for figure 7:

**Source data 1.** PDF file containing original western blots for *Figure 7*, indicating the relevant bands and treatments.

**Source data 2.** Original files for western blots analysis displayed in *Figure 7*.

**Figure supplement 1.** Purified original recombinant proteins (without site-directed acetylation modification) without Kac modifications, validated by Western blot.

**Figure supplement 1—source data 1.** PDF file containing original western blots for *Figure 7—figure supplement 1*, indicating the relevant bands and treatments.

**Figure supplement 1—source data 2.** Original files for western blots analysis displayed in *Figure 7—figure supplement 1*.

**Figure supplement 2.** Characteristics of the overexpressed and purified recombinant target proteins.

**Figure supplement 2—source data 1.** PDF file containing original PCR and SDS-PAGE for *Figure 7—figure supplement 2*, indicating the relevant bands.

**Figure supplement 2—source data 2.** Original files for PCR and SDS-PAGE analysis displayed in *Figure 7—figure supplement 2*.

## Discussion

Lysine acetylation has been confirmed to play crucial roles in almost all essential biological functions in both eukaryotic and prokaryotic cells. Normally, reversibility and dynamicity are necessary characteristics of lysine acetylation. The regulatory mechanism of deacetylation has received as much attention as that of acetylation. In different bacterial species, there are hundreds to thousands of protein lysine acetylation modification sites, which are theoretically unlikely to be regulated by one or a few deacetylases, and there must be some unknown deacetylases in bacteria that act on specific protein substrates to maintain the normal physiological function of cells. Therefore, finding new bacterial deacetylases and identifying their modified protein substrates, so as to better understand the molecular mechanism of their regulation, are important in the research on protein acetylation modification regulation.

Here, we reported that the CobQ protein in *A. hydrophila* is an $NAD^+$- and $Zn^{2+}$-independent deacetylase, and it is very different from currently known deacetylases. We were first interested in the function of AhCobQ based on its annotation as a CobQ/CobB/MinD/ParA family protein in the UniProt database. The deacetylation assay suggested that AhCobQ deacetylated Kac-BSA in the same manner as AhCobB. To our surprise, AhCobQ does not share homology with any prokaryotic sirtuin CobB or currently known eukaryotic deacetylases. We then asked why this was the case since both proteins appeared to belong to the same protein family. Actually, CobB has a conflicting naming phenomenon in the UniProt annotation. There are two distinct CobB annotations in the UniProt database: one is the well-known NAD-dependent protein deacylase, such as NPD_ECOLI in *E. coli*, and

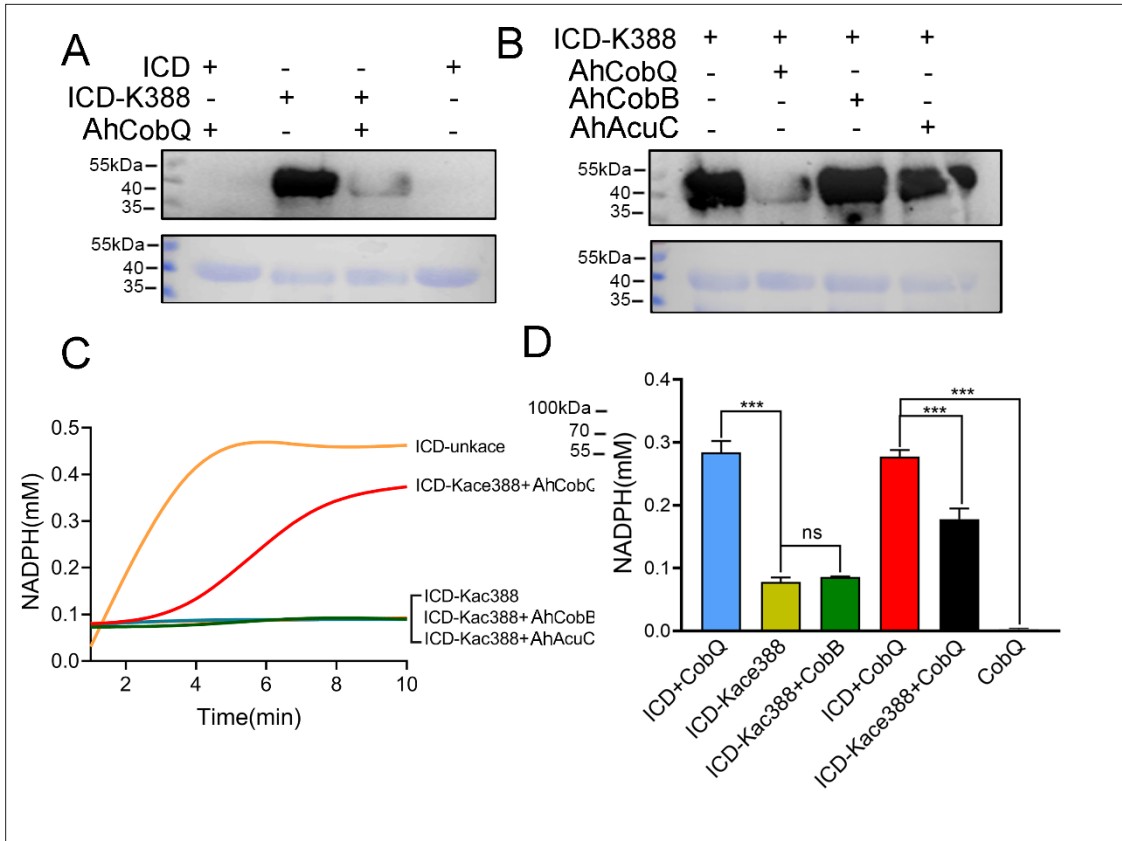

**Figure 8.** AhCobQ positively regulates the isocitrate dehydrogenase (ICD) enzymatic activity through deacetylation of K388 on ICD. (**A**) Western blot verified the deacetylation effect of AhCobQ on ICD and ICD-K388; (**B**) Western blot verified the deacetylation effect of AhCobQ, AhCobB, and AhAcuC on ICD-K388. The PVDF membrane R350 staining for the loading amount control was displayed under the WB results; (**C**) Enzymatic activities of ICD (yellow), ICD-Kac388(blue), ICD-Kac388 treated with AhCobQ (red), ICD-Kac388 treated with AhCobB (green), ICD-Kac388 treated with AhAcuC (brown); (**D**) The histogram showed the effect of AhCobQ/AhcobB on ICD or ICD-Kac388 enzymatic activities at 5 min. ***p<0.001.

The online version of this article includes the following source data for figure 8:

**Source data 1.** PDF file containing original western blots for *Figure 8A and B*, indicating the relevant bands and treatments.

**Source data 2.** Original files for western blots analysis displayed in *Figure 8A and B*.

the other is the ATP-dependent a,c-diamide synthase, such as CobB_SINSX *in Sinorhizobium sp.* that plays an important role in cobalamin biosynthesis (*Galperin and Grishin, 2000*). In some bacterial species, a,c-diamide synthase is also annotated as CbiA (such as CBIA_SALTY, for example), indicating the controversial assignment of CobB. Interestingly, there is not an a,c-diamide synthase homologous protein in *A. hydrophila* ATCC 7966, and the CobB annotation in *A. hydrophila* ATCC 7966 is specific for NAD-dependent protein deacylase. In general, the CobQ protein is annotated as cobyric acid synthase and belongs to a,c-diamide synthase (CobB or CbiA) but not the sirtuin protein (CobB) family. The role of AhCobQ is unknown to date because *A. hydrophila* may import rather than synthesize cobalamin (vitamin B12) (*Seshadri et al., 2006*).

The LC MS/MS analysis of synthesized Kac peptides treated by AhCobQ further validated that this protein is a deacetylase and excluded the possibility of it being a hydrolase. Surprisingly, the enzymatic activity of AhCobB was not affected by $NAD^+$, $Zn^{2+}$, or the sirtuin inhibitor NAM. Given that AhCobQ was reported to be an ATP-dependent cobyric acid synthase (*Galperin and Grishin, 2000*), we also tested the effect of ATP on deacetylase activity but obtained negative results. Thus, the deacetylase activity of AhCobQ was determined to be $NAD^+$-, $Zn^{2+}$-, and ATP-independent. Subsequent assays showed that the deacetylase activity of AhCobQ was mediated by an unknown C-terminal domain in the 195–245 aa range that was downstream of the predicted ATPase domain, suggesting a new domain was responsible for deacetylation. However, the intrinsic enzymatic mechanism needs to be further investigated.

We then further asked whether the protein substrates of AhCobQ were the same as the unknown bacterial KDAC homologs in *A. hydrophila*, AhCobB, and AhAcuC. Although the CobB substrates were identified in *E. coli*, the AcuC substrates and their comparison with the CobB substrates at a high throughput level had not been determined (*Kuhn et al., 2014*; *Weinert et al., 2017*). The characteristics of Kac sites regulated by CobB and AcuC are not yet known. Here, we compared the differential expression of Kac proteins and sites among *ahcobQ*, *ahcobB*, and *ahacuC* mutant and wild-type strains using a quantitative proteomics method. We only focused on the high abundance of Kac proteins or sites (fold change >1.5) when comparing the mutant and WT strains. The results showed that 9.9% of the acetylated proteins (312/3164 Kac protein) were regulated by *ahcobB*, which is consistent with the proportion in the *E. coli cobB* mutant and the previous conclusion that CobB is the predominate deacetylase in some bacterial species (*Weinert et al., 2013*; *AbouElfetouh et al., 2015*). We also found 47 and 186 significantly enriched Kac proteins in Δ*ahcobQ* and Δ*ahacuC*, respectively. Interestingly, there were unique and overlapping Kac substrates among three KDACs, indicating that these KDACs may construct a complicated network for the co-regulation of acetylation proteins. Furthermore, although the enriched GO terms and related substrates were different, all the KDACs appeared to be involved in metabolic pathways.

To further validate our proteomics results, plasmid-based site-specific acetylation was introduced as a target of the recombinant proteins, and the deacetylase activity of the KDACs was determined in vitro. The behavior of the three KDACs on 4 target site-specific acetylated proteins was consistent with our proteomics results. AhCobQ regulated certain proteins at specific Kac sites, which implied that it is involved in specific biological functions. Moreover, AhCobQ and AhCobB also co-regulated the same Kac sites on ArcA-2 (Arginine deiminase), indicating that the cooperation of these KDACs may play an important role during the maintenance of cellular acetylation homeostasis. Most importantly, our results also showed that AhCobQ deacetylated different Kac sites on the same protein as AhCobB or AhAcuC. For example, the bacterial topoisomerase DNA gyrase GyrB is an essential enzyme that controls the topological state of DNA during replication (*Tari et al., 2013*). Previous studies reported the important role of GyrB in bacterial fluoroquinolone resistance (*Chauffour et al., 2021*; *Li et al., 2022*). The 3D structure prediction showed that GyrB K331 is located in the ATPase domain that is responsible for supercoiling DNA, whereas K449 is located in the Toprim domain that is required for the interaction with GyrA (*Brino et al., 2000*). Our current data showed that K331 and K449 were deacetylated by AhCobQ and AhCobB, respectively, indicating the delicate regulatory mechanisms in both functional domains. To our knowledge, this is the first evidence that different KDACs regulate the Kac level at specific sites of the same protein and that they may be involved in differential biological functions in prokaryotic cells. To our knowledge, it is the first evidence that different KADCs regulates the Kac level at specific-site of one protein and that may involve in differential biological functions in prokaryotic cell. We then further wondered the role of AhCobQ in bacterial biological functions. ICD is a key enzyme in the tricarboxylic acid cycle pathway, and also affects bacterial biofilm formation and virulence in several species (*Venkat et al., 2018*). Diverse Kac sites of bacterial ICD have been well identified. In *E. coli* K12, there are at least 15 Kac sites on ICD, and the Kac at K55 and K350 have been determined to play crucial roles on the enzymatic activity by a positively regulatory manner, while Kac at K100, K230, and K235 in a negatively regulatory manner. However, only four Kac sites including K55, K142, K177, and K350 have been proved to be deacetylated by CobB sirtuin obviously. In our quantitative MS data, there are at least ten Kac sites in ICD protein, while only the Kac level of ICD K388 was AhCobQ regulated and others were not altered in KADCs deleted strains. The following assay confirmed that the enzymatic activity of ICD was negatively regulated in a dose-dependent manner by deaectylated ICD K388 site by AhCobQ, whereas not affected by AhCobB and AhAcuC. These results indicate the important role of AhCobQ in bacterial metabolic regulation.

## Conclusion

In summary, here, we identified a novel NAD$^+$- and Zn$^{2+}$-independent deacetylase, AhCobQ, in prokaryotic cells, which does not share homology with current known deacetylases and is unique in prokaryotes. The deacetylase activity of this KDAC is through an unknown domain. Moreover, AhCobQ acetylates at least 47 protein substrates involved in many biological processes, especially metabolic pathways. We provide evidences that AhCobQ negatively regulates the enzymatic activity of bacterial isocitrate dehydrogenase at K388 site. These results indicate that AhCobQ has an important

regulatory role in bacterial metabolic pathways. AhCobQ is a novel deacetylase in prokaryotic cells and its biological functions need to be further investigated.

## Chemicals and reagents

Ni-NTA agarose beads were purchased from Yeasen Biotechnology; C18 ZipTips from Millipore Corporation; water and acetonitrile from Thermo Fisher Scientific; and formic acid (FA), dithiothreitol, acetone, and iodoacetamide from Sigma Aldrich. Synthesized peptides were purchased from Science Peptide (Shanghai, China) with 95–99% purity. All chemicals and reagents used in this study are listed in *Supplementary file 1c*. Critical commercial assay kits used in this study are listed in *Supplementary file 1d*.

## Methods

### Bacterial strains and growth conditions

All strains and vectors involved in this study were preserved or constructed in our laboratory; these are listed in *Supplementary file 1e*. *A. hydrophila* was cultured at 30 °C, and *E. coli* was cultured at 37 °C. All bacterial solutions were transferred at 1:100 (v:v; bacterial solution: LB) and cultured continuously for 16 hr at 200 rpm, unless otherwise specified.

### Generation of gene-deleted strains

According to the whole genome sequence of the *A. hydrophila* ATCC 7966 strain, upstream and downstream fragments of the target gene of about 500 bp were amplified. The amplified gene fragments were ligated with the digested pRE112 plasmid using the restriction sites *SacI* and *XbaI* and then transformed first into *E. coli* MC1061. The plasmid was collected and transformed into *E. coli* S17 and then introduced into *A. hydrophila* by bacterial conjugation. The positive clones were selected on resistant LB plates (100 µg/mL ampicillin and 30 µg/mL chloramphenicol). Finally, the positive mutants from the previous step were spread on LB plates containing 20% sucrose, and a single clone was picked. The positive result was sent to Shanghai Sangon Biological Co., Ltd. for sequencing, and the bacteria were kept at −80 °C for later use (*Li et al., 2019*; *Kou et al., 2022*).

### Physiological phenotype assay

The physiological phenotyping was performed as previously described (*Li et al., 2019*; *Cai et al., 2019*). Briefly, for hemolytic and swarming motility assays, bacteria cultured overnight were spotted on 0.7% sheep blood or 0.4% agar plates, respectively, at 30 °C for 16 hr, and the diameter of the hemolytic or swarming circle was observed and measured in three independent experiments. The bacterial biofilm formation ability was measured with a SpectraMax i3 multifunctional microplate reader using the crystal violet staining method, and the absorbance at an optical density (OD) of 595 nm wavelength was measured, as previously described (*Li et al., 2019*). All these experiments were independently repeated three times, and the one-way ANOVA method was used for statistical analysis.

### BSA acetylation in vitro

Briefly, 1 mg of BSA powder was dissolved in 1 mL of ddH$_2$O containing 0.1 M Na$_2$CO$_3$ and 0.04 g of acetic anhydride and then incubated at 37 °C to react for 3 hr to produce Kac-BSA. Finally, 20 µL of 1 M Tris (pH 8.0) was added to stop the reaction (*Guan et al., 2010*).

### Protein deacetylation assay in vitro

Briefly, 1 µg of protein or peptides were mixed with 0.2 µg of purified KDAC protein in basic buffer (150 mM Tris-HCl, pH 8.0, and 10% glycerol) with 10 mM ZnCl$_2$, 0.5 mM ATP, 0.25 mM NAD$^+$, or 0.5 mM NAM, as appropriate, and then incubated at 37 °C for 3 hr (*Li et al., 2024*).

### Quantitative acetylome analysis

The quantitative acetylome was performed as previously described (*Zhang et al., 2022*). Briefly, the protein samples from bacteria cultured overnight were collected by centrifugation and then sonicated in an ice bath. A 10 mg protein sample was reduced by 10 mM dithiothreitol (DTT), alkylated

by 20 mM iodoacetamide (IAA), and then digested to peptides by trypsin at a 1:50 ratio at 37 °C for 16 hr. The lysine acetylation peptides were enriched by an immuno-affinity enrichment method using an agarose-conjugated anti-lysine-acetylated antibody (Hangzhou Jingjie, PTM 101).

## LC MS/MS

For BSA acetylation identification, the digested peptides were analyzed on an Orbitrap Q Exactive HF mass spectrometer (Thermo Fisher Scientific) equipped with an EASY-nLC 1200 system (Thermo Fisher Scientific). The peptides were first loaded into a C18 trap column (1.9 μm, 100 μm, 20 mm) and separated on an EASY-nLC 1200 UPLC system. The precursor ions were selected for fragmentation in the higher energy collision-activated dissociation (HCD) cell at a normalized collision energy of 27% and then detected using the Orbitrap analyzer at a resolution of 15,000 at m/z 200.

The synthetic peptides YNGVFQECCQAEDKGACLLPK (STD) and YNGVFQECCQA EDK$^{ac}$GA-CLLPK (STD$^{*Kac}$) were analyzed using an Orbitrap Fusion mass spectrometer (Thermo Fisher Scientific) equipped with an EASY-nLC 1200 system (Thermo Fisher Scientific). For the quantitative acetylome, the enriched peptides were analyzed using an Orbitrap Exploris 480 mass spectrometer equipped with a high-field asymmetric waveform ion mobility (FAIMS) Pro spectrometer and an EASY-nLC 1200 system (Thermo Fisher Scientific).

The data obtained were compared and annotated with the BSA (UniProt ID P02769) or *A. hydrophila* ATCC 7966 database from the UniProt database using MaxQuant v1.6.2.10 software. The mass error of the search was set to 10 ppm, the main search was set to 5 ppm, the fragment ion was set to 0.02 Da, and the variable modification was acetylation modification. The maximum false discovery rate (FDR) threshold for proteins, peptides, and modification sites was set at 1%. In addition, the localization probability threshold was specified as >0.75, and the modification site score threshold was set to 40. The precursor area for quantification of synthetic peptides YNGVFQEC CQAEDKGACLLPK (STD) and YNGVFQECCQAEDK$^{ac}$GACLLPK (STD$^{*Kac}$) was manually exacted with Thermo Xcalibur (v.3.0.63). When performing a label-free quantitative (LFQ) comparison, in addition to the above settings, label-free quantification was also set to LFQ, and the intensity signal ratio of the corresponding modified peptides in the identification results (ratio) was >1.5. p<0.05 (Student's *t*-test) among the three biological replicates in each group was considered a significant difference.

## Western blot

The protein samples (0.2 μg) were separated on a 12% sodium dodecyl-sulfate polyacrylamide gel electrophoresis (SDS-PAGE) gel and then transferred to polyvinylidene fluoride (PVDF) membranes using a semi-dry transfer system (Bio-Rad, USA) at 25 V for 20 min. The membranes were blocked with Tris-buffered saline with 0.1% Tween 20 (TBST) containing 5% skim milk for 1 hr and incubated with the diluted primary antibody (PTM-101 1:1,000) for 5 hr at room temperature. After washing three times with TBST, the membranes were incubated with the secondary antibody, horseradish peroxidase-conjugated goat anti-mouse IgG (1:5,000), at room temperature for 1 hr. Finally, the membranes were covered with Clarity western ECL (Bio-Rad) chromogenic solution and exposed with a ChemiDoc MP (Bio-Rad, Hercules, CA, USA). The exposed PVDF membranes were stained with Coomassie R-350 to verify whether the loading amounts were consistent (*Peng et al., 2015*).

## Dot blot

10 mg protein was diluted and spotted on the PVDF membrane. Once the samples had air-dried, the membranes underwent a blocking process with Tris-buffered saline with Tween 20 (TBST) containing 5% BSA for a duration of 3–5 hr. Following this, they were incubated with the PTM-101 antibody at a 1:1000 dilution for 5 hr. Afterward, the membranes were washed three times in TBST, then incubated with a secondary antibody, goat anti-mouse HRP-IgG, at a 1:5000 dilution and containing 3% BSA, for 1 hr at room temperature. This was followed by five washes in TBST, each lasting 5 min. To conclude the process, the membrane was treated with Clarity Western ECL Substrate (Bio-Rad) chromogenic solution, exposed using a ChemiDoc MP imaging system (Bio-Rad, Hercules, CA, USA), and finally stained with Coomassie R-350 to ensure consistent sample loading.

## Recombinant protein production and purification

The target genes were amplified and ligated to the digested pET-21b/32 a plasmid using the restriction sites *EcoRI* and *HindIII*. Then, the recombinant plasmid was transformed into competent *E. coli* BL21 (DE3), and the positive clones were selected from an agar plate with 100 µg/mL ampicillin. For the construction of site-directed mutant strains, the target gene was mutated using a Fast Mutagenesis System kit according to the manufacturer's protocol (Transgen Biotech Co., Beijing, China). All primer pairs used in this study are listed in *Supplementary file 1f*.

The successfully constructed recombinant strains were transferred to 200 mL of LB medium containing 100 µg/mL ampicillin, and then, 200 µL of 1 M isopropyl β-D-1-thiogalactopyranoside (IPTG) was added until the OD at 600 nm reached about 0.6. After washing three times with phosphate-buffered saline (PBS) and dissolving in 10 mL of a solution (5 mM imidazole, 0.5 mM NaCl, 0.05 M Tris-HCl pH = 8.0), the bacterial cells were ultrasonically disrupted for 20 min on ice. The supernatant was collected by centrifuging at 10,000 rpm at 4 °C for 15 min, and the recombinant proteins were purified using an Ni$^+$-NTA column as previously described (*Jiang et al., 2022*).

## Expression and purification of site-specific Kac proteins

We used two genetic code expansion systems in *E. coli* BL21 to express site-specific acetylated proteins (*Vidali et al., 1968* ; *Bryson et al., 2017*). First, the target gene was inserted into the pET-21b plasmid as described above, and the lysine codon at the target position of the gene was substituted with an amber stop (TAG) codon by site-directed mutagenesis. Then, equal amounts of pTECH-MbAcK3RS (IPYE) and recombinant pET-21b plasmid were co-transfected into competent *E. coli* BL21 (DE3) on agar plates containing 100 µg/mL ampicillin and 30 µg/mL chloramphenicol. After culturing for 5 h at 37 °C and 200 rpm, the successful PCR products were confirmed with DNA sequencing.

The successfully site-specific acetylated mutants were cultured overnight in 5 mL of LB medium and then transferred to 200 mL of fresh LB medium containing 100 µg/mL ampicillin and 30 µg/mL chloramphenicol and incubated at 200 rpm and 37 °C. To express the acetyl proteins, 2 mM N-acetyllysine and 20 mM NAM were added to the medium at 37 °C and vortexed at 200 rpm until the OD at 600 nm reached about 0.8. Then, 200 µL of 1 M IPTG was added to the medium, and the cells were cultured at 16 °C and vortexed at 200 rpm for 18 h to express the proteins. The procedure of site-specific Kac protein purification was the same as the method described above.

## Bioinformatics and data analysis

Amino acid sequences were obtained from the UniProt database (https://www.uniprot.org/uniprot). The pfam database (https://www.pfam.org/) was used to predict protein domains, and the online software iTOL (https://itol.embl.de/) was used to generate graphs. The amino acid sequence homology analysis was carried out using position-specific iterated BLAST (PSI-BLAST) software after two iterations (https://blast.ncbi.nlm.nih.gov/Blast.cgi). MEGA v7.0.26 software was used to construct the evolutionary tree that was visualized by online Chiplot software (https://www.chiplot.online/). The online software STRING v.11.5 (http://string-db.org/) was used to perform Kyoto Encyclopedia of Genes and Genomes (KEGG) metabolic pathway enrichment analysis for upregulated proteins, and these proteins were finally visualized using Cytoscape v.3.9.0. Except where otherwise stated, each experiment in this study was independently repeated three times, and the one-way ANOVA method was used for statistical analysis.

## ICD enzymatic activity assay

The enzymatic activity of ICD was detected by measuring the amount of NADPH generated by NADP substrate. 10 µg purified ICD or site-specifically Kac ICD protein was mixed with or without 3 µg purified AhCobQ in 180 µL solution buffer (100 mM Tris-HCl, pH 8.0, 0.5 mM NADP, 3 mM MgCl$_2$, 10% glycerol, 1.5 mM isocitrate), and the absorbance of NADPH at OD340nm was detected by SpectraMax i3(Molecular Devices) once a minute. Simultaneously, serial concentrations of NADPH (0–0.8 mM) was incubated with ICD for standard curve determination.

## Acknowledgements

This work was supported by grants from Key projects of National Natural Science Foundation of China (NSFC) (32171435, 32001045), Natural Science Foundation of Fujian Province (2020J02023), Key Laboratory of Marine Biotechnology of Fujian Province (2020MB04), Natural Science Foundation of Fujian Province (2018J01598), Program for Innovative Research Team in Fujian Agricultural and Forestry University (712018009), and the Fujian-Taiwan Joint Innovative Center for Germplasm Resources and Cultivation of Crop (FJ 2011 Program, 2015–75). We thank the help from Central Laboratory, Fujian Medical University Union Hospital, PR China, and LetPub (https://www.letpub.com/) for its linguistic assistance during the preparation of this manuscript.

## Additional information

### Funding

| Funder | Grant reference number | Author |
|---|---|---|
| National Natural Science Foundation of China | 32171435 | Xiangmin Lin |
| National Natural Science Foundation of China | 32001045 | Xiangmin Lin |
| Natural Science Foundation of Fujian Province | 2020J02023 | Xiangmin Lin |
| Key Laboratory of Marine Biotechnology of Fujian Province | 2020MB04 | Xiangmin Lin |
| Natural Science Foundation of Fujian Province | 2018J01598 | Xiangmin Lin |
| Program for Innovative Research Team in Fujian Agricultural and Forestry University | 712018009 | Xiangmin Lin |
| Fujian-Taiwan Joint Innovative Center for Germplasm Resources and Cultivation of Crop | FJ 2011 Program | Xiangmin Lin |
| Fujian-Taiwan Joint Innovative Center for Germplasm Resources and Cultivation of Crop | 2015–75 | Xiangmin Lin |

The funders had no role in study design, data collection and interpretation, or the decision to submit the work for publication.

### Author contributions

Yuqian Wang, Guibin Wang, Conceptualization, Validation, Writing – original draft, Writing – review and editing; Lishan Zhang, Data curation, Methodology, Writing – review and editing; Qilan Cai, Meizhen Lin, Dongping Huang, Yuyue Xie, Formal analysis; Wenxiong Lin, Resources, Supervision, Funding acquisition; Xiangmin Lin, Resources, Supervision, Funding acquisition, Writing – original draft, Writing – review and editing

### Author ORCIDs

Yuqian Wang ⓘ https://orcid.org/0000-0002-9216-9553
Xiangmin Lin ⓘ https://orcid.org/0000-0003-2216-0198

Reviewer #1 (Public review): https://doi.org/10.7554/eLife.97511.4.sa1
Reviewer #2 (Public review): https://doi.org/10.7554/eLife.97511.4.sa2
Author response https://doi.org/10.7554/eLife.97511.4.sa3

## Additional files

### Supplementary files
Supplementary file 1. LC MS/MS of acetylated peptides and materials used in this study. (a) Selected Kac peptide quantification among Kac-BSA and Kac-BSA incubated with CobQ and BSA without acetylation, by LC MS/MS. (b) Acetylation-modified upregulated proteins of AhCobQ-deleted strains and the identification of residue positions by LC MS/MS. (c) Chemicals and reagents used in this study. (d) Critical commercial assay kits used in this study. (e) Bacterial strains and plasmids used in this study. (f) Primer pairs used in this study.

MDAR checklist

### Data availability
The raw MS data were deposited in the public Integrated Proteome Resources (iProx) database with the dataset identifier IPX0005366000.

The following dataset was generated:

| Author(s) | Year | Dataset title | Dataset URL | Database and Identifier |
|---|---|---|---|---|
| Lin X | 2022 | Aeromonas hydrophila CobQ is a new type of NAD+/Zn2+ independent protein lysine deacetylase in prokaryote | https://www.iprox.cn//page/project.html?id=IPX0005366000 | iProX, IPX0005366000 |

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
