## [Editor Report · eLife Assessment]

In this **valuable** study, the authors studied a novel Zn2+- and NAD+-independent KDAC protein, AhCobQ, in Aeromonas hydrophila, which lacks homology with eukaryotic counterparts, thus underscoring its unique evolutionary trajectory within the bacterial domain. They attempt to demonstrate deacetylase activity, however, whilst the revised manuscript has been improved, significant aspects of the data are still **incomplete** and require further refinement. The work will be of interest to microbiologists studying metabolism and post-translational modifications.

---

## [Referee Report · Reviewer #1 (Public review)]

Summary:

This study by Wang et al. identifies a new type of deacetylase, CobQ, in Aeromonas hydrophila. Notably, the identification of this deacetylase reveals a lack of homology with eukaryotic counterparts, thus underscoring its unique evolutionary trajectory within the bacterial domain.

Strengths:

The manuscript convincingly illustrates CobQ's deacetylase activity through robust in vitro experiments, establishing its distinctiveness from known prokaryotic deacetylases. Additionally, the authors elucidate CobQ's potential cooperation with other deacetylases in vivo to regulate bacterial cellular processes. Furthermore, the study highlights CobQ's significance in the regulation of acetylation within prokaryotic cells.

Weaknesses:

The problem I raised has been well resolved. I have no further questions.

---

## [Referee Report · Reviewer #2 (Public review)]

In recent years, lots of researchers tried to explore the existence of new acetyltransferase and deacetylase by using specific antibody enrichment technologies and high resolution mass spectrometry. Here is an example for this effort. Yuqian Wang et al. studied a novel Zn2+- and NAD+-independent KDAC protein, AhCobQ, in Aeromonas hydrophila. They studied the biological function of AhCobQ by using biochemistry method and MS identification technology to confirm it. These results extended our understanding of the regulatory mechanism of bacterial lysine acetylation modifications. However, I find this conclusion is a little speculative, and unfortunately it also doesn't totally support the conclusion as the authors provided.

Major concerns:

-It is a little arbitrary to come to the title "Aeromonas hydrophila CobQ is a new type of NAD+- and Zn2+-independent protein lysine deacetylase in prokaryotes." It should be modified to delete the "in the prokaryotes" except that the authors get new more evidence in the other prokaryotes for the existence of the AhCobQ.

-I was confused about the arrangement of the supplementary results. Because there are no citations for Figures S9-S19.

-Same to the above, there are no data about Tables S1-S6.

-All the load control is not integrated. Please provide all of the load controls with whole PAGE gel or whole membrane western blot results. Without these whole results, it is not convincing to come the conclusion as the authors mentioned in the context.

-Thoroughly review the materials & methods section. It is unclear to me what exactly the authors describe in the method. All the experimental designs and protocols should be described in detail, including growth conditions, assay conditions, and purification conditions, etc.

-Include relevant information about the experiments performed in the figure legends, such as experimental conditions, replicates, etc. Often it is not clear what was done based on the figure legend description.

---

## [Author Response]

The following is the authors’ response to the current reviews.

**Public Reviews:**

**Reviewer #1 (Public review):**
Summary:This study by Wang et al. identifies a new type of deacetylase, CobQ, in Aeromonas hydrophila. Notably, the identification of this deacetylase reveals a lack of homology with eukaryotic counterparts, thus underscoring its unique evolutionary trajectory within the bacterial domain.Strengths:The manuscript convincingly illustrates CobQ's deacetylase activity through robust in vitro experiments, establishing its distinctiveness from known prokaryotic deacetylases. Additionally, the authors elucidate CobQ's potential cooperation with other deacetylases in vivo to regulate bacterial cellular processes. Furthermore, the study highlights CobQ's significance in the regulation of acetylation within prokaryotic cells.Weaknesses:The problem I raised has been well resolved. I have no further questions.

Thanks for your valuable comments very much.

**Reviewer #2 (Public review):**
In recent years, lots of researchers tried to explore the existence of new acetyltransferase and deacetylase by using specific antibody enrichment technologies and high resolution mass spectrometry. Here is an example for this effort. Yuqian Wang et al. studied a novel Zn2+- and NAD+-independent KDAC protein, AhCobQ, in Aeromonas hydrophila. They studied the biological function of AhCobQ by using biochemistry method and MS identification technology to confirm it. These results extended our understanding of the regulatory mechanism of bacterial lysine acetylation modifications. However, I find this conclusion is a little speculative, and unfortunately it also doesn't totally support the conclusion as the authors provided.Major concerns:- It is a little arbitrary to come to the title "Aeromonas hydrophila CobQ is a new type of NAD+- and Zn2+-independent protein lysine deacetylase in prokaryotes." It should be modified to delete the "in the prokaryotes" except that the authors get new more evidence in the other prokaryotes for the existence of the AhCobQ.

Thank you for your suggestion. However, I believe there has been some confusion regarding the title. In the revised manuscript we have already updated the title to: "Aeromonas hydrophila CobQ is a new type of NAD+- and Zn2+-independent protein lysine deacetylase."

This title does not include the phrase "in prokaryotes," as you mentioned. We kindly suggest verifying the version of the manuscript that was reviewed to ensure you are reviewing the most recent changes.

- I was confused about the arrangement of the supplementary results. Because there are no citations for Figures S9-S19.

Thank you for your feedback. It appears there may have been a misunderstanding, possibly due to reviewing an outdated version of the manuscript. In the revised manuscript we revised the supplementary figures and now have only 12 figures, all of which are correctly cited in the manuscript on pages 12 to 15. Below is a detailed list of the updated figure citations:

Figures S1: page 8, line 148;

Figures S2: page 9, line 168;

Figures S3 and S4: page 10, line 178;

Figures S5: page 10, line 186;

Figures S6: page 10, line 189;

Figures S7: page 12, line 221;

Figures S8-S10: page 13, line 245;

Figures S11: page 11, line 282;

Figures S12: page 15, line 286

- Same to the above, there are no data about Tables S1-S6.

Thank you for your attention to the supplementary materials. As with the figures, we have already uploaded the data for Tables S1-S6 in the revised manuscript on November 19, 2024, and properly cited Tables S1 – S6 in the manuscript. Below is the citation information:

Tables S1: page 10, line 194;

Tables S2: page 13, line 245;

Tables S3: page 21, line 438;

Tables S4: page 22, line 439;

Tables S5: page 22, line 445;

Tables S6: page 27, line 564.

Please note that Tables S3 – S4 include the chemical reagents, primers, and other experimental materials, which are not intended to be cited in the results section.

- All the load control is not integrated. Please provide all of the load controls with whole PAGE gel or whole membrane western blot results. Without these whole results, it is not convincing to come the conclusion as the authors mentioned in the context.

Thank you for your comment. Please note that the full membrane western blot results were included in the revised manuscript. We hope this satisfies your request. If you need further clarification or additional data, please do not hesitate to let us know.

- Thoroughly review the materials & methods section. It is unclear to me what exactly the authors describe in the method. All the experimental designs and protocols should be described in detail, including growth conditions, assay conditions, and purification conditions, etc.

Thank you for your valuable suggestion. In response to your comment and previous feedback, we have alredy revised the Materials & Methods section thoroughly in the revised manuscript. The experimental details, including growth conditions, assay protocols, and purification procedures, are described in full on pages 22 to 30 of the revised manuscript.

- Include relevant information about the experiments performed in the figure legends, such as experimental conditions, replicates, etc. Often it is not clear what was done based on the figure legend description.

Thank you very much for your detailed feedback and suggestions. We have made sure to describe what each data point represents in the figure legends, as per the previous feedback. However, we would like to clarify that while we have provided detailed descriptions in the legends, the inclusion of every specific experimental condition in the figure legends could result in redundancy, as these details are already thoroughly outlined in the Materials & Methods section.

We hope this explanation addresses your concern.

**Recommendations for the authors:**

**Reviewer #1 (Recommendations for the authors):**
I have no further revision comments.

Thank you very much.

**Reviewer #2 (Recommendations for the authors):**
I carefully read the point-to-point response from the author. Although they listed lots of the reasons for the ugly results, it still can not persuade me to accept their conclusions. While, as I know, it is impossible to reject their work in eLife as it was sent out for peer-review. I also can't accuse them of being wrong, but I have my opinion on this point. That is not the results, but the attitude.

Thank you for your feedback. However, I must express some concerns regarding the nature of your comments. Based on the issues you've raised, it seems that you may have reviewed an outdated version of the manuscript. In the updated revision we addressed all the points you've raised, including the figure and table citations, experimental methods, and data integration.

We understand that differing opinions are part of the peer-review process, but we respectfully believe that your conclusion regarding our attitude is based on a misunderstanding, possibly caused by reviewing an incorrect version of the manuscript. We have always strived to approach this manuscript with utmost professionalism and have diligently responded to each of your concerns.

We sincerely suggest reviewing the latest version of our manuscript, and we welcome any further constructive feedback. We hope this clarifies any misunderstandings and look forward to your continued support.

Thank you for your time and thoughtful consideration.

The following is the authors’ response to the original reviews.

**Public Reviews:**

**Reviewer #1 (Public review):**
Summary:This study by Wang et al. identifies a new type of deacetylase, CobQ, in Aeromonas hydrophila. Notably, the identification of this deacetylase reveals a lack of homology with eukaryotic counterparts, thus underscoring its unique evolutionary trajectory within the bacterial domain.Strengths:The manuscript convincingly illustrates CobQ's deacetylase activity through robust in vitro experiments, establishing its distinctiveness from known prokaryotic deacetylases. Additionally, the authors elucidate CobQ's potential cooperation with other deacetylases in vivo to regulate bacterial cellular processes. Furthermore, the study highlights CobQ's significance in the regulation of acetylation within prokaryotic cells.Weaknesses:The problem I raised has been well resolved. I have no further questions.
**Reviewer #2 (Public review):**
In recent years, lots of researchers tried to explore the existence of new acetyltransferase and deacetylase by using specific antibody enrichment technologies and high resolution mass spectrometry. Here is an example for this effort. Yuqian Wang et al. studied a novel Zn2+- and NAD+-independent KDAC protein, AhCobQ, in Aeromonas hydrophila. They studied the biological function of AhCobQ by using biochemistry method and MS identification technology to confirm it. These results extended our understanding of the regulatory mechanism of bacterial lysine acetylation modifications. However, I find this conclusion is a little speculative, and unfortunately, it also doesn't totally support the conclusion as the authors provided.
**Reviewer #3 (Public review):**
Summary:This study reports on a novel NAD+ and Zn2+-independent protein lysine deacetylase (KDAC) in Aeromonas hydrophila, termed as AhCobQ (AHA_1389). This protein is annotated as a CobQ/CobB/MinD/ParA family protein and does not show similarity with known NAD+-dependent or Zn2+-dependent KDACs. The authors showed that AhCobQ has NAD+ and Zn2+-independent deacetylase activity with acetylated BSA by western blot and MS analyses. They also provided evidence that the 195-245 aa region of AhCobQ is responsible for the deacetylase activity, which is conserved in some marine prokaryotes and has no similarity with eukaryotic proteins. They identified target proteins of AhCobQ deacetylase by proteomic analysis and verified the deacetylase activity using site-specific Kac proteins. Finally, they showed that AhCobQ activates isocitrate dehydrogenase by deacetylation at K388.Strengths:The finding of a new type of KDAC has a valuable impact on the field of protein acetylation. The characters (NAD+ and Zn2+-independent deacetylase activity in an unknown domain) shown in this study are very unexpected.Weaknesses:(1) The characters (NAD+ and Zn2+-independent deacetylase activity in an unknown domain) shown in this study are very unexpected. To convince readers, MSMS data must be necessary to accurately detect (de)acetylation at the target site in the deacetylase activity assay. The authors showed the MSMS data in assays with acetylated BSA, but other assays only rely on western blot.(2) They prepared site-specific Kac proteins and used them in deacetylase activity assays. Incorporation of acetyllysine at the target site should be confirmed by MSMS and shown as supplementary data.(3) The authors imply that the 195-245 aa region of AhCobQ may represent a new domain responsible for deacetylase activity. The feature of the region would be of interest but is not sufficiently described in Figure 5. The amino acid sequence alignments with representative proteins with conserved residues would be informative. It would be also informative if the modeled structure predicted by AlphaFold is shown and the structural similarity with known deacetylases is discussed.
**Recommendations for the authors:**

**Reviewer #1 (Recommendations for the authors):**
The problem I raised has been well resolved. I have no further questions.
**Reviewer #2 (Recommendations for the authors):**
Questions to response of"-The load control is not all integrated. All of the load controls with whole PAGE gel or whole membrane western blot results should be provided. Without these whole results, it is not convincing to come to the conclusion that the authors have."Just as the Authors answered. The Coomassie Blue R-350 staining outcomes from the PVDF membranes. That is a good control for the experiment. However, I still have several questions about it:(1) The first is the quality of these Western blot. Why all the bands of these Western blot is so ugly? To tell the truth, it is very difficult to come to a conclusion from these poor western blots.

We appreciate your feedback regarding the quality of the Western blots presented in Figure 7. We believe the “ugly bands” you referred to reflect our results validating the functions of CobQ through the use of recombinant site-specific Kac protein substrates.

In our study, we meticulously engineered these recombinant site-specific Kac proteins using a two-plasmid system, based on foundational research published in Nature Chemical Biology (2017, 13(12): 1253-1260), which introduced the genetic encoding of Nε-acetyllysine into recombinant proteins. However, we faced a common challenge: protein truncation due to premature translation termination at the reassigned codon. This issue not only hampers protein yields, as discussed in ChemBioChem (2017, 18(20): 1973-1983), but also contributes to the suboptimal appearance of the Western blot results.

Despite conducting at least two independent repetitions for the Western blot analysis of the site-specific Kac proteins, which yielded consistent results, we recognize that the overall quality remains less than ideal. This variability is inherently related to the characteristics of the target proteins. Nevertheless, the primary aim of our manuscript is to validate the novel deacetylase activity of CobQ. We have provided multiple lines of evidence, including mass spectrometry (MS/MS) and Western blot analyses, to substantiate this claim. In response to your comments, we have decided to remove the ambiguous Western blot results from Figure 7, retaining only four figures that demonstrate significant differences across at least two independent replicates (Author response images 1-5). Additionally, we have included four biological replicates of the Western blot results for ICD Kac388 + CobQ in the supplementary materials (Author response image 5) to further validate the deacetylase function of CobQ.

**Author response image 1. sa3fig1:** Western blot validation of the Kac26 AcrA-2 protein substrates regulated by the three KDACs in two biological replicates.

**Author response image 2. sa3fig2:** Western blot validation of the Kac48 Sun protein substrates regulated by the three KDACs in two biological replicates.

**Author response image 3. sa3fig3:** Western blot validation of the Kac103 Sun protein substrates regulated by the three KDACs in two biological replicates.

**Author response image 4. sa3fig4:** Western blot validation of the Kac195 Eno protein substrates regulated by the three KDACs in three biological replicates.

**Author response image 5. sa3fig5:** Western blot validation of the Kac388 ICD protein substrates regulated by AhCobQ in this study. Each sample was independently repeated at least three time.

(2) The second is why some of the results are not from the same PVDF by comparing the Coommassie staining with the WB results just as authors responded. For example, the HrpA-K816 (ac), Eno-K195 (ac), ArcA-2-K26 (ac), ArcA-2-K26 (ac), IscS-K93(ac), A0KJ75-K81(ac), GyrB-K331(ac), GyrB-K449(ac), FtsA-K320(ac), FtsA-K409(ac), RecA-K279(ac), and the RecA-K306(ac). All of them are clearly not from the same staining results of PVDF membrane but from a new PVDF membrane.

We assure you that the R-350 stained PVDF membranes originate from the same Western blot membranes. However, we acknowledge that visual discrepancies may arise due to differences in imaging techniques. The Western blot results were scanned using a ChemiDoc MP (Bio-Rad, Hercules, CA, USA), while the Coomassie R-350 stained PVDF membranes were captured using a standard camera. These differences can create a misleading appearance, making it seem as though they come from different membranes.

It is also important to note that the intensity of the protein marker cannot be directly compared between the two imaging methods. As illustrated in Author response image 6, the protein marker at 70 kDa is clearly detectable in the Coomassie R-350 image, whereas it may not be as apparent in the Western blot result due to inherent differences in detection sensitivity.

**Author response image 6. sa3fig6:** The comparison of Western blotting and R-350 strained results of same protein marker in the same PVDF membrane. The protein marker located at 70 kDa can be detected easily in Coomassie R-350, while is difficult to display in WB result.

Additionally, we have removed some of the so-called "ugly" Western blot results in the updated manuscript and provided the original full film of the relevant images as an attachment. This documentation demonstrates that all the data you referenced originate from the same film, as shown in Figures 1-5.

(3) The third is why there is no replication for all these WB results. We should draw a conclusion with serious attitude, but not from the only one repeat, even say nothing about the poor results.

Thank you for your valuable suggestion. In the second version of the manuscript, we have included the original full film of the relevant images. While we previously explained the reasons behind the "ugly" Western blot results, we have decided to remove some, or even all, of these results from Figure 7 in the updated version. The related images will be updated in the supplementary materials (Figures 1-5 in responding letter and Figure 7 in the revised manuscript).

Furthermore, we have provided a more detailed discussion regarding the poor results in the updated manuscript to ensure clarity and transparency. We appreciate your understanding and hope these changes meet your expectations.

Questions to response of " L174-187, L795 (Please show the whole membrane (or PAGE gel) of the loading control of CobB, and CobQ, except for the Kac-BSA)".(1) As we all educated that there is no control, and no biology. Where is the band of CobQ? Why do not stain the same PVDF membrane with R-350 staining but with a new membrane?

Thank you for your insightful feedback. As noted in our previous response, the absence of visible bands for AhCobQ and AhCobB on the Coomassie R-350 stained PVDF membrane is primarily due to the low loading amounts and protein loss during the Western blotting process.

To reinforce our findings, we repeated the analysis of the protein samples via SDS-PAGE, using the same loading quantity as in the previous Western blot shown in Figure 2 of the manuscript. As illustrated in Author response image 7, the bands for CobB and CobQ are discernible, albeit with significantly lower intensities compared to the Kac-BSA bands. Upon examining the full Coomassie R-350 stained PVDF membranes provided in Supplementary Material 1, we observe that the CobB and CobQ bands are not easily visible. This aligns with your observations and can be attributed to potential protein loss during the transfer from SDS-PAGE to the PVDF membrane.

**Author response image 7. sa3fig7:** The SDS-PAGE gel displayed the loading amounts of Kac-BSA and CobB/CobQ.

To enhance the visibility of the CobQ/CobB bands, we increased the loading of CobQ/CobB in a new Western blot experiment, using 2 µg of Kac-BSA in combination with 0.8 µg of CobQ/CobB. As shown in Figure 8, while the increasing amounts of Kac-BSA resulted in a more blurred signal, the bands for the recombinant CobQ and CobB proteins were clearly detectable. This indicates that both proteins were indeed involved in the in vitro protein deacetylation assay.

**Author response image 8. sa3fig8:** Western blot verified the deacetylase activity assay of AhCobQ and AhCobB on Kac-BSA.

Furthermore, we conducted a mass spectrometry analysis comparing Kac-BSA and Kac-BSA incubated with CobQ, as well as BSA without acetylation, against the *A. hydrophila* database with a cut-off of unique matched peptides >1. It is challenging to completely avoid contaminant detection during protein purification, especially when using high-resolution mass spectrometry. Our findings revealed that CobQ has the highest number of unique matched peptides (Author response table 1), while contaminants such as AHA_3036, AHA_0497, AHA_1279, and valS could be excluded, as they were present in Kac-BSA or BSA samples. Additionally, Tuf1, RplQ, GroEL, RpsF, RpsU, RpsB, RpsO, and RpsJ are known ribosomal subunits or chaperonins that are abundantly expressed in cells and may interact with various proteins, leading to contaminant detection.

**Author response table 1. sa3table1:** LC MS/MS results of selected peptide quantification among Kac-BSA and Kac-BSA incubated with CobQ and BSA without acetylation against *A. hydrophila* database (unique matched peptides>1).

Gene name	Unique peptide	Description	Kac-BSA+AhCobQ intensity	Kac-BSA intensity	BSA intensity
*cobQ*	11	CobQ/CobB/MinD/ParA family protein	2.59E+09	0	0
*tuf1*	7	Elongation factor Tu	5.35E+09	0	0
*rp/Q*	4	Large ribosomal subunit protein bL17	3.44E+08	0	0
*groEL*	3	Chaperonin GroEL	1.6E+08	0	0
*AHA_3036*	3	VWA domain-containing protein	1.03E+08	23562000	5005500
*rpsF*	2	Small ribosomal subunit protein bS6	79780000	0	0
*rpsU*	2	Small ribosomal subunit protein bS21	56694000	0	0
*rpsB*	2	Small ribosomal subunit protein uS2	47779000	0	0
*rpsO*	2	Small ribosomal subunit protein uS15	15869000	0	0
*rpsJ*	2	Small ribosomal subunit protein uS10	58784000	0	0
*AHA_0497*	2	ATP-grasp domain-containing protein	20711000	50345000	0
*AHA_1279*	2	Major outer membrane protein OmpAll	7316100	18183000	0
*valS*	2	Valine--tRNA ligase	0	173130000	0
*aceE*	2	Pyruvate dehydrogenase E1 component	74854000	0	0

Although AceE, a pyruvate dehydrogenase E1 component, theoretically possesses deacetylase activity, this possibility is low. First, in the SDS-PAGE gel of the purified recombinant protein, CobQ is the major band, with other proteins present at very low levels (less than 1/10 of CobQ). This suggests that significant deacetylation by contaminants is unlikely. Second, we purified His-tagged AhCobQ and GST-fused AhCobQ separately and tested their deacetylase activities. As shown in Figure S4 of the updated manuscript, both purified AhCobQ proteins exhibited deacetylase activity, while the negative control (purified GST protein only) did not, further supporting our conclusion that enzyme activity is not attributable to contaminating proteins (Figure S5).

(2) Without the CobB and CobQ bands, it is impossible to say the function of CobQ is a new deacetylase. To avoid this confusion, it is easy to run a new gel and stain it with anti-His antibody to show these deacetylases.

Thank you very much for your suggestion. We have performed the experiment in the comment (1) as your suggestion.

(3) The explanation about the CobB/CobQ bands are not visible is not acceptable. Because the molecular weight of the CobB and CobQ is smaller than that of BSA, it is impossible that these bands will be loss during membrane transfer.

Thank you for your valuable feedback. I completely agree that the loss of CobB and CobQ proteins during membrane transfer is unlikely due to their smaller molecular weight compared to BSA. As shown in Figure 7, the bands for CobB and CobQ are detectable in the SDS-PAGE gel but not visible on the Coomassie R-350 stained PVDF membrane.

Several factors could contribute to this issue. One possibility is that the detection sensitivity of Coomassie R-350 may be lower than that of Coomassie R-250 used in the gel. Additionally, the Western blot results using an anti-His antibody further indicate low loading amounts of CobB and CobQ proteins on the PVDF membrane (Figure 8). This suggests that the observed low levels may indeed be due to protein loss during the membrane transfer process, despite their relatively small size.

**Reviewer #3 (Recommendations for the authors):**
(1) I found Tables S1 and S2 in the revised manuscript. It is strange to me that the intensity of Kac-BSA+CobQ is zero, completely nothing. Typically, a portion of the acetylated peptide remains after the deacetylation reaction.

Thank you for your observation. When we report an intensity of zero, it does not imply a complete absence of signal; rather, it indicates that the signal for the target peptide is below the detectable threshold. This is likely due to the minimum cut-off setting in the MaxQuant (MQ) software, which is determined by parameters like "peptide_mass_tolerance" (as discussed in MQ user groups online, though it may not be explicitly listed in the parameters file).

In our study, we performed a deacetylase assay that demonstrated CobQ's rapid activity; for instance, it can deacetylate ICD-K388ac within just four minutes. This leads me to hypothesize that the CobQ + Kac-BSA sample may have undergone near-complete enzymatic hydrolysis during the reaction.

Furthermore, Table S1 in manuscript presents only a selection of the mass spectrometry results to illustrate CobQ's activity. In addition to the 15 acetylated peptides shown, there are many more (27 peptides) that exhibit significantly reduced acetylation levels without reaching zero intensity. The overall acetylation level of BSA peptides incubated with CobQ is calculated to be only 0.13 times that of Kac-BSA (Diagnostic peak: yes, peptide score: >100, Localization probability: >0.95) (Author response image 9).

Based on these findings, we believe our mass spectrometry results are reliable and effectively support our conclusions. Thank you for your understanding.

**Author response image 9. sa3fig9:** The intensities of all Kac peptides of Kac-BSA with or without AhCobQ incubation in LC MS/MS.

(2) It would be better to provide the information about ArcA and ArcA-2 as mentioned in the authors' response. It would be helpful for readers to understand that they are different proteins.

Thank you for your suggestion. In the *A. hydrophila* ATCC 7966 dataset, there are indeed two distinct proteins referred to as ArcA: ArcA-1, which functions as an aerobic respiration control protein, and ArcA-2, which acts as an arginine deiminase. Importantly, these two proteins do not share any sequence homology; they are only similarly named due to their acronyms. While we believe this distinction does not require extensive explanation in the current study, we appreciate your input. Additionally, in response to Reviewer 2’s feedback, we have decided to remove the Western blot result for ArcA-2 due to its poor quality in the updated manuscript.

(3) Line 409-416. Despite my comment, the citation of related papers on ICD acetylation in *E. coli* is still missing.

Thank you for your suggestion. It has been added and highlighted in red. (Venkat S, et al, 2018, 430(13): 1901-1911)

(4) The image resolution of Figure 3C and 3D is still bad. I could not evaluate that Kac was exactly incorporated at the target site.

Thank you for your feedback regarding the image resolution of Figures 3C and 3D. We have now displayed these figures with improved clarity, as you suggested.

To further validate the reliability of our MS2 data, we employed Proteome Discoverer 2.4 (Thermo) to analyze the raw data and provide theoretical mass information. As shown in Author response images 10-13, the MS2 spectra and fragment match lists for both unmodified and acetylated peptides offer additional confirmation of the reliability of our mass spectrometry results.

**Author response image 10. sa3fig10:** MS2 spectrum of unmodified peptide using PD v2.4 software.

**Author response image 11. sa3fig11:** The theoretical mass of unmodified peptide by PD 2.4.

**Author response image 12. sa3fig12:** MS2 spectrum of acetylated peptide using PD v2.4 software.

**Author response image 13. sa3fig13:** The theoretical mass of acetylated peptide by PD 2.4.

(5) Again, in Figure 8D, it should be shown the significance between ICD-Kac388 and ICD-Kac388+AhCobB to support the authors' conclusion that AhCobQ activates ICD by deacetylation at K388.

Thanks for your suggestion, we have updated the figure in Figure 8D in updated manuscript.

(6) It was nice that the authors presented the mass spectrum data of ICD-K388 acetylation (Figure 2 in responding letter). However, the data did not convince me that K388 is acetylated. In the figure, two b-ion peaks are detected, 285.1557 and 386.2034, which may correspond to NK (theoretical mass, 260.15) and NKT (theoretical mass, 361.20) peptides, respectively. If K388 is acetylated, an increase in the mass of 42 should be observed, but the difference between the detected and theoretical mass is 25. I also could not understand what the peak of 126.0913 mass is, indicated with acK* in red.

Thank you for your detailed observation. The data presented in the MS2 spectrum for ICD-K388 acetylation in Figure 2 of the previous response letter were generated using Proteome Discoverer 2.4 (PD, Thermo) to ensure accurate mass calculations. Similar to the results from MaxQuant, ICD-K388 was identified again (Author response image 14).

Regarding the b-ion peaks you mentioned, the values 285.1557 and 386.2034 correspond to NK^ac^ and NK^ac^T peptides, respectively. The theoretical masses for these peptides are as follows: NK^ac^ (285.15 = 115.05020 + 128.095 + 42.01) and NK^ac^T (386.20 = NK^ac^ + 101.04768). The differences between the theoretical and detected masses for the relevant b-ions (b2*-NK, b52+-NH3, and b3) are minimal, at 0.00 Da and 2.1 ppm, respectively, which is consistent with the incorporation of an NH3 group (Author response image 15).

**Author response image 14. sa3fig14:** The MS2 of ICD-K388 peptide by PD 2.4.

**Author response image 15. sa3fig15:** The theoretical mass of ICD-K388 peptide by PD 2.4.

The peak at 126.0913 m/z, indicated as acK*, represents immonium ions of ε-N-acetyllysine, which are generated during the fragmentation of acetyllysine. This diagnostic ion is widely recognized as a marker for identifying acetylated peptides (Nakayasu, et al,. A method to determine lysine acetylation stoichiometries. International journal of proteomics. 2014;2014(1):730725; Trelle et al., Utility of immonium ions for assignment of ε-N-acetyllysine-containing peptides by tandem mass spectrometry. Analytical chemistry. 2008;80(9):3422-30). Additionally, it is a default parameter in MaxQuant for identifying Kac peptides (Author response image 16).

Based on these findings, we believe the evidence supporting ICD-K388 acetylation is robust.

**Author response image 16. sa3fig16:** The default parameter in Kac peptide identification in Maxquant v1.6 software.

(7) As mentioned by other reviewers, some of the figures and tables are incomplete. Some panels (ex. Figure 7C and 7D) and explanations (ex. What are lanes 1, 2, and 3 in Figure S3) are still missing.

Thank you for your suggestion. It has been added.